# Physical and functional interaction between A20 and ATG16L1-WD40 domain in the control of intestinal homeostasis

Karolina Slowicka[1,2,3,10], Inmaculada Serramito-Gómez [4,10], Emilio Boada-Romero[4,10], Arne Martens [1,2], Mozes Sze[1,2,3], Ioanna Petta[1,3,5], Hanna K. Vikkula[1,2,3], Riet De Rycke[1,2,6,7], Eef Parthoens[1,2,6], Saskia Lippens[1,2,6], Savvas N. Savvides [1,8], Andy Wullaert[1,3,9], Lars Vereecke[1,3,5,11], Felipe X. Pimentel-Muiños[4,11] & Geert van Loo [1,2,3,11]

Prevention of inflammatory bowel disease (IBD) relies on tight control of inflammatory, cell death and autophagic mechanisms, but how these pathways are integrated at the molecular level is still unclear. Here we show that the anti-inflammatory protein A20 and the critical autophagic mediator Atg16l1 physically interact and synergize to regulate the stability of the intestinal epithelial barrier. A proteomic screen using the WD40 domain of ATG16L1 (WDD) identified A20 as a WDD-interacting protein. Loss of A20 and Atg16l1 in mouse intestinal epithelium induces spontaneous IBD-like pathology, as characterized by severe inflammation and increased intestinal epithelial cell death in both small and large intestine. Mechanistically, absence of A20 promotes Atg16l1 accumulation, while elimination of Atg16l1 or expression of WDD-deficient Atg16l1 stabilizes A20. Collectively our data show that A20 and Atg16l1 cooperatively control intestinal homeostasis by acting at the intersection of inflammatory, autophagy and cell death pathways.

[1] VIB Center for Inflammation Research, 9052 Ghent, Belgium. [2] Department of Biomedical Molecular Biology, Ghent University, 9052 Ghent, Belgium. [3] Ghent Gut Inflammation Group (GGIG), Ghent University, 9000 Ghent, Belgium. [4] Instituto de Biología Molecular y Celular del Cáncer (IBMCC), Centro de Investigación del Cáncer, CSIC-Universidad de Salamanca, 37007 Salamanca, Spain. [5] Department of Rheumatology, Ghent University, 9000 Ghent, Belgium. [6] VIB BioImaging Core Ghent, 9052 Ghent, Belgium. [7] UGent TEM Expertise Centre, Ghent University, 9000 Ghent, Belgium. [8] Department of Biochemistry and Microbiology, Ghent University, 9000 Ghent, Belgium. [9] Department of Internal Medicine and Pediatrics, Ghent University, 9000 Ghent, Belgium. [10]These authors contributed equally: Karolina Slowicka, Inmaculada Serramito-Gómez, Emilio Boada-Romero. [11]These authors jointly supervised this work: Lars Vereecke, Felipe X. Pimentel-Muiños and Geert van Loo. Correspondence and requests for materials should be addressed to F.X.P-Mño. (email: fxp@usal.es) or to G.v.L. (email: geert.vanloo@irc.vib-ugent.be)

nflammatory bowel disease (IBD) is a chronic and debilitating inflammatory pathology of the gut that affects ~1 in 200 people in developed countries and exhibits alarming incidence expansion worldwide[1]. Crohn's disease, which can affect any part of the gastrointestinal tract, and ulcerative colitis, which is restricted to the colon mucosa, are the two main clinical manifestations of IBD[2]. In healthy conditions, the intestinal epithelium maintains a solid physical barrier established by the tight contact of cells preventing bacterial infiltration and subsequent inflammation. Moreover, specialized secretory epithelial cell types such as Paneth and goblet cells provide innate immune defense functions by secreting antibacterial peptides and mucus, limiting bacterial adhesion and infiltration. Hence, IBD is thought to arise in genetically predisposed individuals due to exaggerated responses of the host immune system to intestinal bacteria, and defects in maintaining the stability of the mucosal barrier[3]. Genome wide association studies (GWAS) have identified more than 200 genes associated with IBD, many of which are involved in the regulation of innate immune responses and intestinal epithelial functions, including *A20* and *ATG16L1*[2,4–6].

The intestinal epithelium senses luminal antigens through pattern recognition receptors (PRRs) including toll-like receptors (TLRs), NOD-like receptors (NLRs), RIG-I-like receptors, and C-type lectin receptors, leading to the activation of the nuclear factor κB (NF-κB) pathway[7]. Although generally linked to inflammation, NF-κB activation in intestinal epithelial cells (IECs) is essential for regulating important protective mechanisms including maintaining gut barrier integrity by inducing anti-apoptotic proteins, antimicrobial peptides, and mucins[3,7–9]. However, aberrant NF-κB activation leads to the production of numerous inflammatory mediators, causing chronic inflammation. Indeed, excessive NF-κB activation can be observed in mucosa of IBD patients[10] and in experimental mouse models of colitis[3,7–9]. An important anti-inflammatory and enterocyte protective factor induced by NF-κB is A20 (also known as TNFα-induced protein 3, *TNFAIP3*)[11–13]. A20 acts as a ubiquitin-editing protein that terminates NF-κB and cell death signaling in response to PRR and cytokine receptor stimulation[13]. IEC-specific deletion of A20 was shown to sensitize mice to experimental colitis and TNF toxicity, due to increased epithelial apoptosis, identifying A20 as a crucial barrier protective factor, indispensable for maintaining intestinal barrier integrity during inflammatory stress[12,14]. The physiological importance of A20 in intestinal pathology is further strengthened by the identification of *A20/TNFAIP3* polymorphisms associated with Crohn's disease, ulcerative colitis, and celiac disease[13]. GWAS have also identified polymorphisms in *ATG16L1* and other autophagy-related genes in IBD, suggesting autophagy-dependent mechanisms for controlling intestinal immune homeostasis[4,5,15,16]. ATG16L1 mediates the assembly of a macromolecular complex that lipidates LC3/ATG8 to promote formation of canonical double-membrane autophagosomes[17]. However, ATG16L1 also performs alternative activities that are apparently unrelated to autophagosome generation, including anti-inflammatory functions[18,19]. Mammalian ATG16L1 includes a C-terminal domain formed by 7 WD40-type

repetitions (the WD40 domain, WDD)[20] that is dispensable for the canonical autophagic pathway[21,22]. Instead, this region appears to function as a docking site for adapter proteins that engage ATG16L1 to perform unconventional activities[22–25]. Consistent with this idea, the anti-inflammatory role of ATG16L1 in NOD signaling has been proposed to involve interaction between NOD1/2 and the WDD[19]. Identification of WDD adapter molecules and their associated functions is likely to provide novel insights into how ATG16L1 regulates inflammation and other unconventional activities.

The most common IBD-linked polymorphism in *ATG16L1*, the T300A allele, was shown to prevent binding of the WDD to proteins containing a novel WDD-binding amino acid motif[22], and also to facilitate ATG16L1 processing by caspase 3 leading to defects in anti-bacterial autophagy and enhanced cytokine responses[26,27]. Knock-in mice expressing the Atg16l1 T300A variant also display morphological defects in Paneth and goblet cells[27]. Paneth cell function abnormalities can also be observed in Atg16l1 hypomorphic mice, and in mice deficient for Atg16l1 or the autophagy genes Atg5 and Atg7 in the intestinal epithelium[28–30]. Atg16l1 was recently shown to control barrier integrity by protecting the intestinal epithelium from TNF–induced apoptosis and/or necroptosis in experimental models of colitis[28,31,32]. Prevention of autophagosome formation can also induce the accumulation of immature autophagosomal membranes that promote caspase 8 activation and apoptosis in an LC3-dependent manner[33].

Overall, different cellular mechanisms including PRR-induced inflammatory signaling, autophagy and cell death signaling pathways are closely intertwined to control intestinal immune homeostasis. Defects in any of these mechanisms may have detrimental consequences for the epithelial barrier integrity and predispose to the development of intestinal pathology. However, disease development may require the convergence of multiple defects affecting several layers of control on intestinal immune homeostasis, due to functional redundancy or molecular compensatory counterbalance mechanisms.

In this study, we identify A20 as a direct binding partner of the WDD of ATG16L1, and examine whether a genetic interaction might exist in the context of intestinal homeostasis. We find that A20 restricts Atg16l1 accumulation, and vice versa, to regulate autophagic, inflammatory, and cell death responses. To study their functional relationship in vivo, we generate mice with double A20 and Atg16l1 deficiency in the intestinal epithelial compartment. These A20-Atg16l1$^{\Delta IEC}$ mice develop spontaneous IBD-like pathology characterized by severe inflammation in both small and large intestine, crypt abscesses with marked epithelial cell death, Paneth cell loss, and villi erosion, in contrast to single A20$^{\Delta IEC}$ or Atg16l1$^{\Delta IEC}$ mice, which do not develop overt intestinal abnormalities nor inflammation in homeostatic conditions. Together, our data indicate that inflammatory, cell death and autophagy signaling converge at the level of A20 and Atg16l1 to maintain intestinal immune homeostasis, while each protein can compensate for the other's loss to some extent.

## Results

**Proteomic identification of ATG16L1 WDD-binding proteins.** In order to characterize the physiological function of the ATG16L1 WDD, we developed a proteomics approach to identify proteins able to interact with this region. This approach was based on expression of GST-WDD chimeric proteins in cultured cells and identification of co-precipitating proteins by mass spectrometry. Initial optimization studies showed that a GST-ATG16L1 chimera containing the WDD as annotated in sequence databases (residues 320–607) showed lower expression

**Table 1 Total number of non-redundant proteins identified per cell line and condition**

| Cells/conditions | Number |
| --- | --- |
| JAR | 364 |
| THP-1 (untreated) | 531 |
| THP-1 (LPS/PMA) | 417 |
| Total proteins identified: 1044. | |

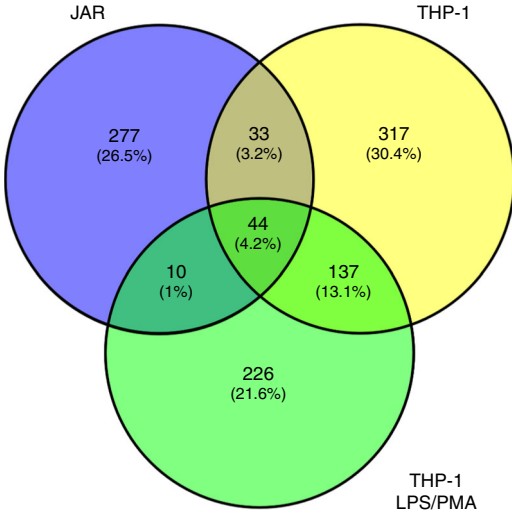

**Fig. 1** Limited overlapping of WDD-interacting proteins identified in the different cellular systems. Venn diagram showing the degree of overlapping between the collection of candidates found in the three cellular systems analyzed (see Supplementary Data 3 for more complete information)

| Table 2 Number of identified proteins belonging to the indicated functional families | |
| --- | --- |
| **Protein families** | |
| **Process** | **Number** |
| Inflammation/innate immunity | 54 |
| Apoptosis/cell death | 35 |
| Adhesion | 45 |
| Autophagy | 22 |
| DNA replication/damage/repair | 41 |
| Immune response | 56 |
| Cell signaling | 186 |
| Intracellular trafficking | 27 |

| Table 3 Selection of proteins that play a direct role in the regulation of inflammatory and/or innate immune signaling pathways (Uniprot annotation) | |
| --- | --- |
| **Selected proteins (inflammation/innate immunity)** | |
| NALP2/NLRP2 | IKBKG/NEMO |
| NALP4/NLRP4 | PRKCD |
| TNFAIP3/A20 | TRAFD1 |
| CYLD | STING |
| TNIP | TRIM25 |
| IFIH1/MDA5 | RIOK3 |
| TRIM29 | ISG15 |

Supplementary Data 2) that were not substantially overlapping between the three cellular systems analyzed (Fig. 1; Supplementary Data 3), suggesting that the WDD interactome has considerable cell type and developmental state specificity. The identified proteins are involved in a wide variety of biological processes (Table 2; Supplementary Data 4), pointing to a broad functional diversification of the WDD. Interestingly, we found a number of proteins previously linked to the regulation of innate immunity and inflammatory signaling (as annotated in the Uniprot database; Table 3; Supplementary Data 4). Among these are NALP2 and NALP4 (members of the NLRP family of intracellular innate immune receptors), the cytoplasmic viral sensor MDA5 and the anti-inflammatory protein A20/TNFAIP3 (Table 3). Co-immunoprecipitation studies upon co-expression of the relevant partners in HEK-293T cells confirmed that NALP2, MDA5, and A20 are able to interact with the WDD of ATG16L1 (residues 320–607; Fig. 2a–c), but not with an N-terminal region (1–299, Fig. 2a–c) that suffices to sustain canonical autophagy[22]. These results suggest that the WDD may participate in innate immune and inflammatory signaling by interacting with mediators involved in these pathways.

**A20 binds ATG16L1 through its N-terminal OTU domain.** Since polymorphisms in both A20 and ATG16L1 are associated with IBD, we further characterized the interaction between these two proteins. Co-immunoprecipitation assays between GST-ATG16L1 fusion constructs and deleted versions of A20 showed that the N-terminal region of A20 (amino acids 1–263) is sufficient to bind full-length ATG16L1 (Fig. 3a). The WDD of ATG16L1 (residues 320–607) mediates this interaction in both HEK293T (Supplementary Fig. 2A) and intestinal HCT116 (Supplementary Fig. 2B) cells. Residues 92–263 containing the OTU domain that harbors the deubiquitinating activity was the minimal region of A20 able to bind the WDD (Fig. 3b). Interestingly, the interaction between ATG16L1 and A20 is not altered by the T300A allele that increases the risk for Crohn's disease (Supplementary Fig. 3A). Previous results showed that the WDD recognizes an amino acid motif comprising [YFW]-X-X-L as the critical positions, an element originally identified in the intracellular region of the transmembrane molecule TMEM59[35]. We found 7 versions of an inclusive form of this motif ([YFW]-X-X-[LVI]) in the 92–263 domain of A20. Inactivation of each individual motif by mutation of both essential residues to alanine did not inhibit co-precipitation between the WDD and A20–1–263 (Supplementary Fig. 3B), but simultaneous mutation of all motifs impaired such interaction (Supplementary Fig. 3C). These results suggest that the different WDD-binding motifs present in the N-terminal region of A20 cooperatively participate in the recognition of the WDD. Two nonsynonymous SNP's in the OTU domain (A125V and F127C) have been linked to increased risk for IBD and autoimmune diseases[36–38], but when introduced in the A20-1-263 construct did not appear to influence its interaction with the WDD (Supplementary Fig. 3D). Treatment with TNF promoted co-precipitation between endogenous A20 and ATG16L1 in Atg16L1-deficient mouse embryonic fibroblasts (MEFs) and HCT116 intestinal epithelial cells restored with HA-ATG16L1 (Fig. 3c), indicating that both molecules interact in response to physiological stimuli.

**Loss of A20 increases Atg16l1 and LC3-II levels.** Recent studies have demonstrated that autophagy pathways get activated in inflammatory conditions as a cellular defense mechanism in order to protect against the harmful effects of inflammatory reactions[16]. To study the effect of A20 deficiency on inflammatory signaling

levels compared to a longer version that includes amino acids 231–607 (Supplementary Fig. 1A), consistent with the recently published structure of the WDD showing that position 320 interrupts a beta strand that might be part of the domain and important for its stability[34]. Therefore, we focused on expressing GST-ATG16L1-231-607 in JAR (choriocarcinoma) and THP-1 (monocytic) cells for subsequent proteomic studies. The latter were activated with LPS/PMA to favor expression of inflammatory mediators or left untreated. From these assays, we found a total of 1044 proteins (Table 1; Supplementary Data 1;

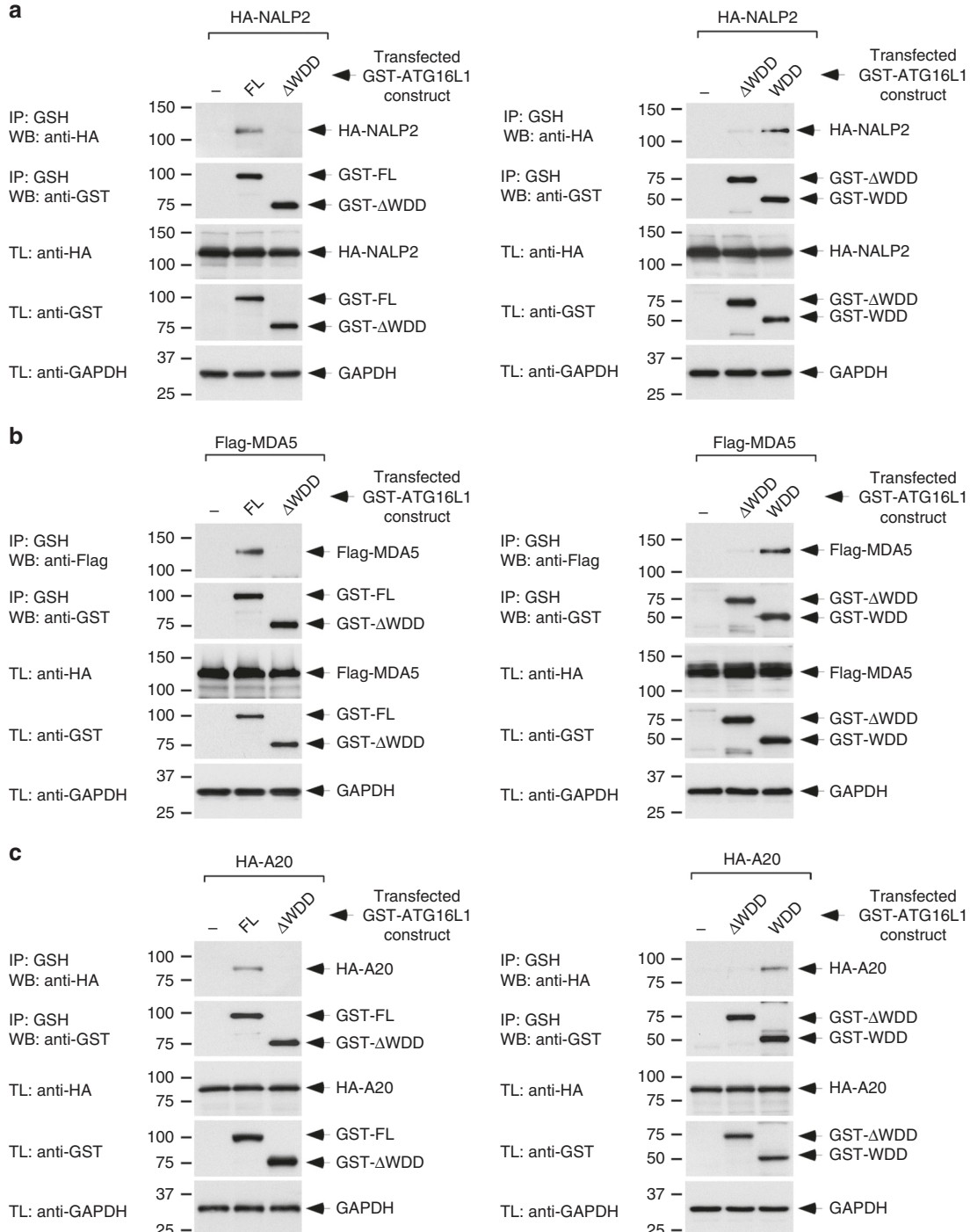

**Fig. 2** NALP2, MDA5, and A20 bind the WDD of ATG16L1 in co-precipitation assays. **a–c** HEK-293T cells were transfected with the indicated constructs (WDD, residues 320–607 of ATG16L1; ΔWDD, 1–299), lysed 36 h after transfection and subjected to GST immunoprecipitation using agarose beads coupled to glutathione (IP: immunoprecipitation; TL: total lysate). Shown are Western-blots against the indicated molecules

and autophagy, we evaluated the expression of autophagy markers after TNF stimulation in A20 wild-type and A20 deficient MEFs. As previously reported, A20 deficient cells show prolonged phosphorylation and sustained degradation of the NF-κB inhibitory molecule IκBα, consistent with the importance of A20 as a negative feedback regulator of inducible NF-κB activation (Fig. 4a). In addition to the enhanced activation of the NF-κB pathway, Atg16l1 expression levels are increased in A20-deficient cells, and more microtubule-associated protein 1 light chain 3 (LC3) protein associates with phosphatidylethanolamine (LC3-

II), both in basal conditions and upon TNF treatment (Fig. 4a and Supplementary Fig. 4A). Also p62, a multifaceted scaffolding protein involved in trafficking proteins to autophagy (and itself a substrate for autophagic degradation), is slightly induced in A20 deficient MEFs (Fig. 4a), suggesting reduced autophagic flux. However, accumulation of LC3-II in the absence of A20 persisted after lysosomal inhibition with bafilomycin (Supplementary Fig. 5A, B), arguing that it reflects enhanced autophagic flux in A20-deficient cells. Interestingly, ectopic expression of the WDD in A20-deficient cells inhibited LC3-II induction by TNF in a

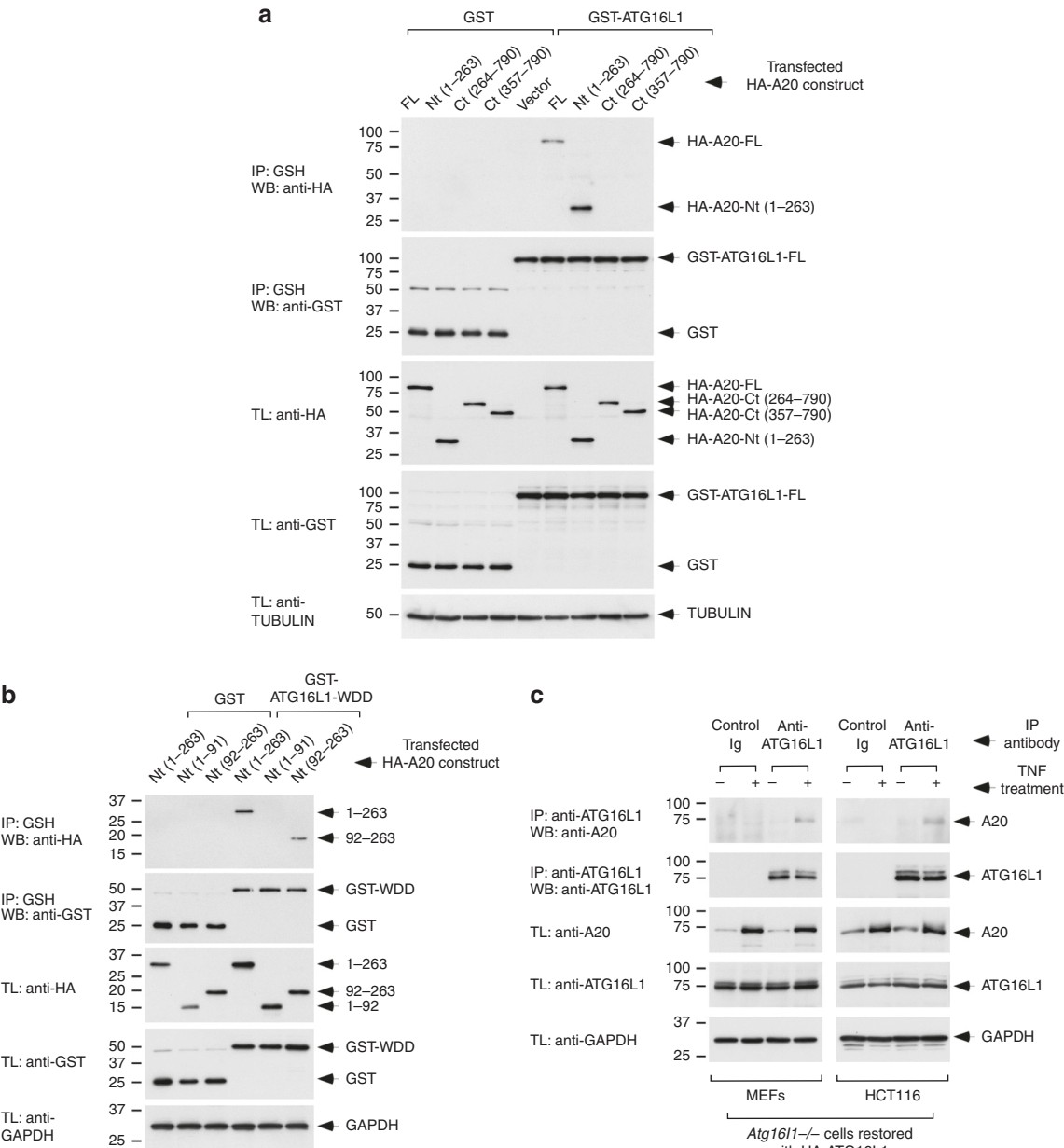

**Fig. 3** ATG16L1 recognizes the N-terminal region of A20 (92–263) through the WDD. **a** A stretch of A20 including amino acids 1–263 suffices to co-precipitate with GST-ATG16L1. **b** A minimal region between amino acids 92–263 of A20 suffices to interact with the WDD of ATG16L1. Co-immunoprecipitation assays were done as in Fig. 2. **c** Endogenous A20 coprecipitates with ATG16L1 upon TNF treatment of *Atg16L1*[−/−] MEFs and HCT116 cells restored with HA-ATG16L1. The indicated cells were treated with 30 ng/ml of TNF for 2 h, and processed for anti-ATG16L1 immunoprecipitation and western-blotting with the shown antibodies

dominant-negative manner (Supplementary Fig. 5C), suggesting that the autophagic response has unconventional, WDD-mediated features that might help explain the apparently contradicting behavior of LC3-II and p62 in this setting. Alternatively, p62 is an NF-κB response gene which can be strongly induced in absence of A20[39]. Atg16l1 expression is also induced in small intestinal organoids from A20 deficient mice, particularly in response to TNF (Fig. 4b and Supplementary Fig. 4B). No difference in *Atg16l1* expression could be measured between both genotypes at the transcript level, concluding that the effect of A20 on Atg16l1 expression is regulated at the protein level (Fig. 4c, d). A20 has been shown to regulate the stability of NF-κB signaling proteins, including RIPK1, through ligation of K48-linked

polyubiquitin chains and subsequent proteasomal degradation[40]. However, we have no evidence that Atg16l1 is ubiquitinated and/or stabilized upon inhibition of the proteasome (Supplementary Fig. 6), so the molecular mechanism causing enhanced Atg16l1 expression in absence of A20 remains elusive.

**Atg16l1 regulates A20 expression.** To explore a possible role of the WDD in the interplay between A20 and Atg16l1, we reconstituted *Atg16l1/A20*-double deficient MEFs with HA-A20 and/or different ATG16l1 constructs. In these assays we were unable to recapitulate the impact of A20-deficiency on ATG16L1 expression (Fig. 4e, f), perhaps because the subtle effect observed for endogenous Atg16l1 (Fig. 4a) is lost when

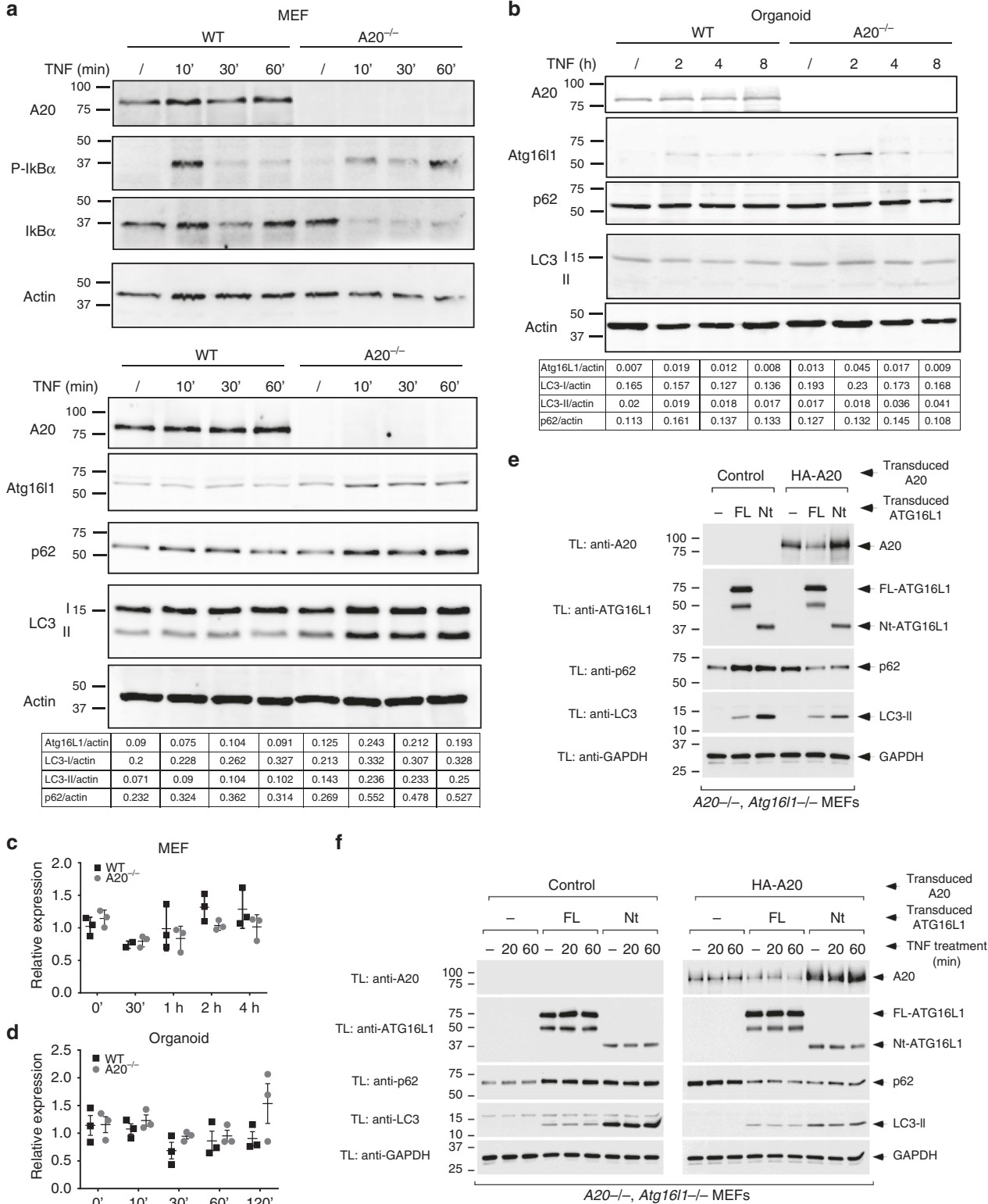

**Fig. 4** A20 deficiency increases Atg16l1 expression and LC3-II expression levels. **a** Immortalized MEFs were stimulated with 1000 IU/ml of recombinant murine TNF for indicated time points. Data representative of five independent experiments. **b** Small intestinal organoids were stimulated with 10 ng/ml recombinant TNF for 2, 4, and 8 h. Data representative of three independent experiments. **c**, **d** Atg16l1 mRNA expression in MEFs (**c**) and organoids (**d**) from wild-type (WT) and A20$^{\Delta IEC}$ (A20$^{-/-}$) mice either or not stimulated with TNF for indicated time points. Data are presented as mean±SD and are representative of two independent experiments. **e**, **f** A20 is destabilized by ATG16L1 through the WDD. **e** A20/Atg16l1-double deficient MEFs were retrovirally transduced first with HA-A20 and subsequently with the indicated ATG16L1 constructs (Nt refers to the same ΔWDD ATG16l1 construct (aminoacids 1–299) used in Fig. 2). Cells were lysed 5 days after the last transduction and the resulting total cell lysates were processed for western-blotting with the indicated antibodies. **f** The same cellular strains as in **e** were treated with TNF (30 ng/ml) for the indicated times and processed for western-blotting against the indicated molecules

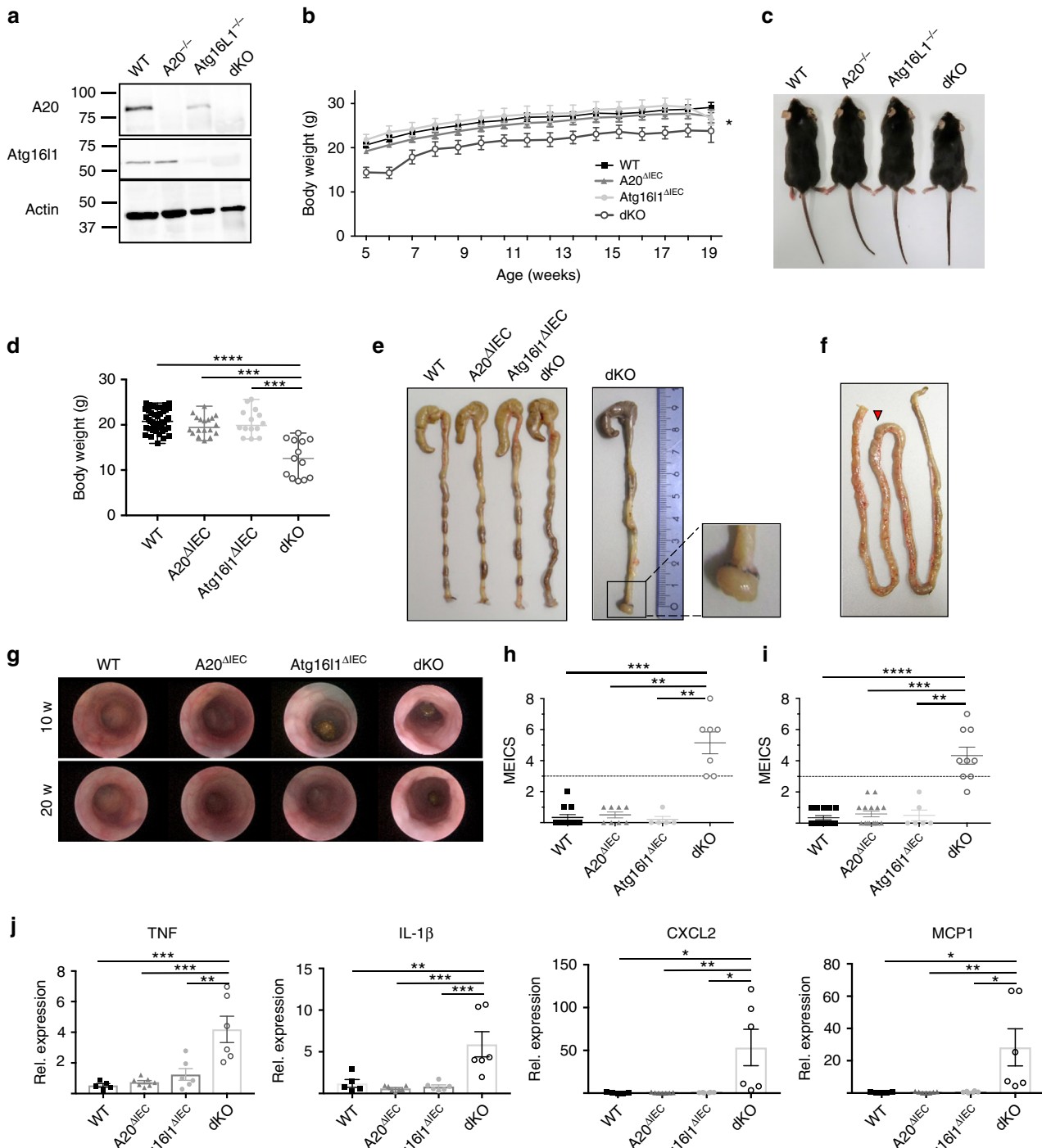

**Fig. 5** IEC-specific A20/Atg16l1 deficient mice develop spontaneous intestinal pathology. **a** Representative Western blot analysis of small intestinal organoid cultures from wild-type (WT), A20$^{\Delta IEC}$, Atg16l1$^{\Delta IEC}$, and A20/Atg16l1 dKO mice. **b** Absolute body weight in grams of WT ($n = 18$), A20$^{\Delta IEC}$ ($n = 10$), Atg16l1$^{\Delta IEC}$ ($n = 7$), and littermate A20/Atg16l1 dKO ($n = 8$) mice in function of time. Data are expressed as mean ± SEM. *$p < 0.05$ for dKO mice compared to the three control groups, as analysed as repeated measurements using the residual maximum likelihood as implemented in Genstat v17. **c** Representative picture of 20-week-old littermate WT, A20$^{\Delta IEC}$, Atg16l1$^{\Delta IEC}$, and A20/Atg16l1 dKO mice. **d** Body weight of WT, A20$^{\Delta IEC}$, Atg16l1$^{\Delta IEC}$, and A20/Atg16l1 dKO mice at the age of 5 weeks. Each symbol represents one mouse. Data represent mean±SEM. ****$p < 0.0001$; ***$p < 0.001$, as analyzed in GraphPad Prism 7 with Kruskal-Wallis test. **e** Representative macroscopic colon images of 20-week-old WT, A20$^{\Delta IEC}$, Atg16l1$^{\Delta IEC}$, and dKO littermate mice showing rectal prolapses and colon thickening in dKO animals. **f** Representative macroscopic image of the small intestine of a 20-week-old dKO mouse showing increased vascularization and swelling in the jejunum (red arrow head). **g** Representative endoscopic pictures of 10-week old WT, A20$^{\Delta IEC}$, Atg16l1$^{\Delta IEC}$, and dKO mice demonstrating spontaneous colon pathology in dKO mice. **h**, **i** Colon inflammation detected by endoscopy and represented as mouse endoscopic index of colitis severity (MEICS) score in 10-week-old (**h**) and 20-week-old (**i**) WT, A20$^{\Delta IEC}$, Atg16l1$^{\Delta IEC}$, and dKO mice. Each symbol represents one mouse. Data represent mean±SEM. ****$p < 0.0001$; ***$p < 0.001$; **$p < 0.01$, as analyzed in GraphPad Prism 7 with Kruskal-Wallis test. **j** Quantitative PCR for TNF, IL-1β, CXCL2, and MCP1 in small intestinal lysates from WT, A20$^{\Delta IEC}$, Atg16l1$^{\Delta IEC}$, and dKO mice. Each symbol represents one mouse. Data represent mean±SEM. ***$p < 0.001$; **$p < 0.01$; *$p < 0.05$, as analysed in GraphPad Prism 7 with One-way ANOVA

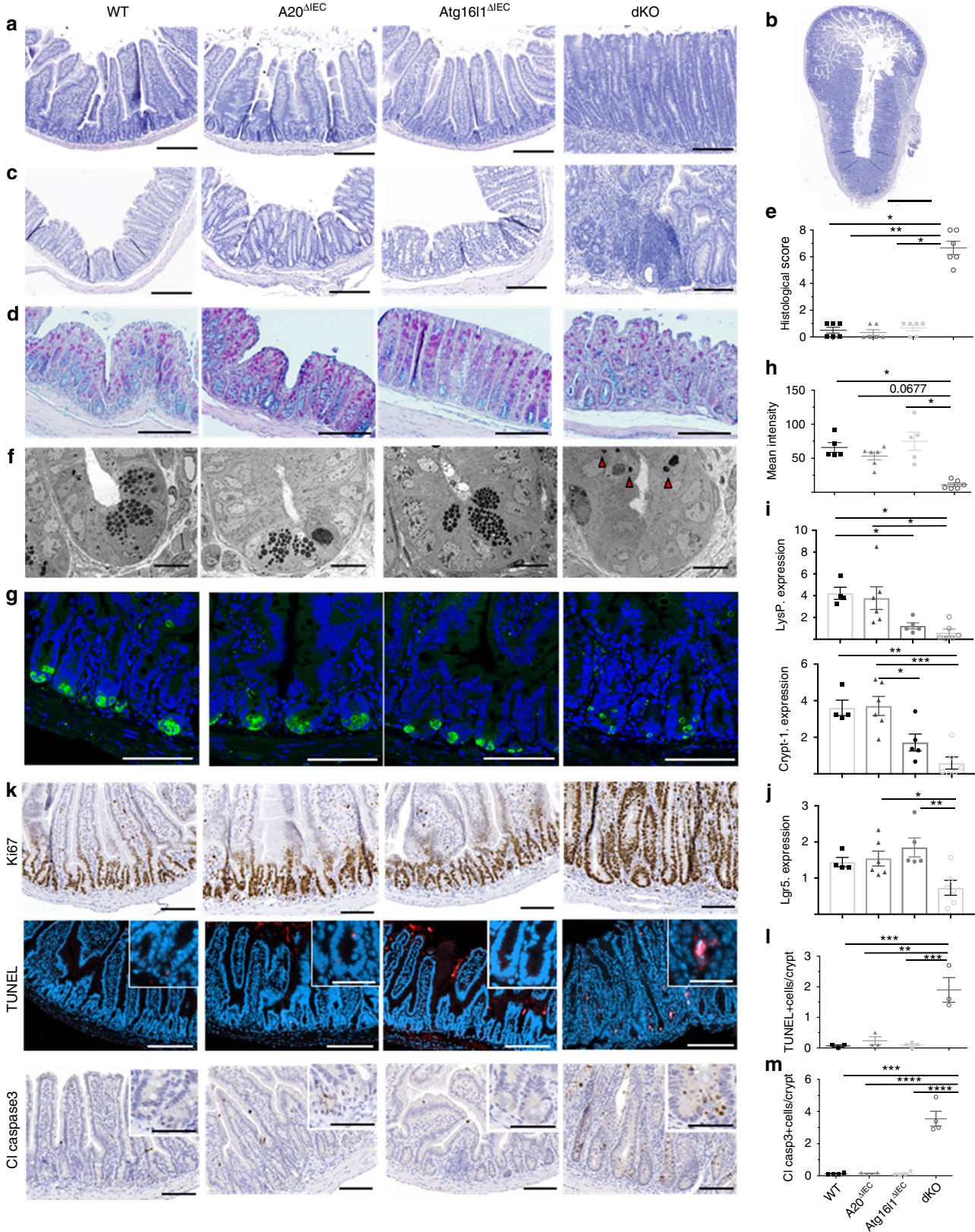

both contenders are expressed at higher levels. However, we found reduced A20 expression levels in the presence of full-length ATG16L1 (Fig. 4e), particularly upon TNF treatment (Fig. 4f). Cells reconstituted with the N-terminal region of ATG16L1 (residues 1–299 that suffice to sustain autophagy) showed recovered A20 expression (Fig. 4e, f), indicating that

ATG16L1 downregulates A20 through the WDD. Both A20 and ATG16L1 are expressed from constitutive retroviral transgenes in this system, so the observed changes in A20 abundance are likely to be post-transcriptional. Consistently, treatment with lysosomal inhibitors (bafilomycin, E64d/pepstatin) tended to normalize A20 expression levels (Supplementary Fig. 7A),

**Fig. 6** Intestinal inflammation, Paneth cell loss and crypt cell apoptosis in IEC-specific A20/Atg16l1 deficient mice. **a, b** Hematoxylin-eosin (H&E) staining of proximal ileum sections of 20-week-old control (WT), A20$^{\Delta IEC}$, Atg16l1$^{\Delta IEC}$, and dKO mice (**a**; scale bar, 200 μm), with an overview of transmural inflammation and ulceration present in the dKO (**b**; scale bar, 1000 μm). **c** H&E staining of colon sections of 20 week old WT, A20$^{\Delta IEC}$, Atg16l1$^{\Delta IEC}$, and dKO mice. Scale bar, 200 μm. **d** AB/PAS staining of colon sections from WT, A20$^{\Delta IEC}$, Atg16l1$^{\Delta IEC}$ and dKO mice. Scale bar, 200 μm. Note extensive inflammation and crypt loss in dKO sections. **e** Histological scoring of small intestinal sections from WT, A20$^{\Delta IEC}$, Atg16l1$^{\Delta IEC}$, and dKO mice. Each symbol represents one mouse. Data represent mean±SEM. **$p < 0.01$; *$p < 0.05$, as analyzed in GraphPad Prism 7 with Kruskal–Wallis test. Images representative of $n = 6$ mice per genotype. **f** Transmission electron (TEM) micrographs of WT, A20$^{\Delta IEC}$, Atg16l1$^{\Delta IEC}$, and dKO mice. Scale bar 10 μm. Note Paneth cell loss and presence of condensed apoptotic cell bodies (indicated by arrow heads) in dKO sections. Representative images for $n = 3$ for each genotype. **g** Immunofluorescent staining of jejunal sections using an antibody recognizing lysozyme in intracellular granules of Paneth cells (green) in WT, A20$^{\Delta IEC}$, Atg16l1$^{\Delta IEC}$, and dKO mice. Cell nuclei were counterstained with DAPI (blue). Scale bar, 100 μm. **h** Quantification of lysozyme intensity. Each symbol represents one mouse. Data represent mean ± SEM.*$p < 0.05$, as analyzed in GraphPad Prism 7 with Kruskal–Wallis test. **i, j** Quantitative PCR analysis for Paneth cell Lysozyme (LysP), cryptdin-1 (crypt-1) (**i**) and stem cell Lgr5 (**j**) expression in small intestinal lysates from WT, A20$^{\Delta IEC}$, Atg16l1$^{\Delta IEC}$, and dKO mice at 20 weeks of age. Each symbol represents one mouse. Data represent mean±SEM. ***$p < 0.001$; **$p < 0.01$; *$p < 0.05$, as analysed in GraphPad Prism 7 with One-way ANOVA. **k** Immunostaining for Ki67, TUNEL (red) and cleaved caspase 3 on sections from the small intestine of WT, A20$^{\Delta IEC}$, Atg16l1$^{\Delta IEC}$, and dKO mice. Images representative of $n = 5$ mice per genotype. Cell nuclei were counterstained with DAPI. Scale bars, bright-field 100 μm; fluorescence 200 μm; inserts 50 μm. **l, m** Quantification of TUNEL (**l**) and cleaved caspase 3-positive cells (**m**) in sections from the small intestine of WT, A20$^{\Delta IEC}$, Atg16l1$^{\Delta IEC}$, and dKO mice. Data represent mean±SEM. ****$p < 0.0001$; ***$p < 0.001$; **$p < 0.01$, as analysed in GraphPad Prism 7 with One-way ANOVA

arguing that a lysosomal pathway (perhaps autophagy) degrades A20 more intensely in the presence of full-length ATG16L1. While the N-terminal domain of ATG16L1 fully restored the basal autophagic flux in A20-positive cells (p62 and LC3 blots in Fig. 4e, f), its LC3 lipidating activity was deregulated in cells lacking A20 expression (increased levels of LC3-II but no impact on p62 clearance; Fig. 4e, f), thus revealing a functional epistasis between A20 and the WDD in the control of LC3 lipidation. More generally, ATG16L1 only restored the basal autophagic flux in cells harboring A20 (see p62 Western-blots in Fig. 4e, f), pointing to a wider function of A20 in the regulation of autophagy. To examine the impact of ATG16L1 WDD-dependent degradation of A20 in NF-κB activation, we measured NF-κB-dependent luciferase activity induced by TNF in all restored cell lines. In these assays we found that the WDD of ATG16L1 regulates NF-κB activation in cells lacking A20, because cells expressing FL-ATG16L1 were less responsive than those devoid of ATG16L1 expression or those harboring just the N-terminal domain (Supplementary Fig. 7B). All strains restored with HA-A20 showed poor NF-κB activation irrespective of their ATG16L1 status, likely because A20 over-expression dominantly inhibits the activity.

In an extension of these studies, we confirmed that endogenous A20 is also upregulated in TNF-treated Atg16l1-deficient MEFs compared to their wild-type counterparts, with the appearance of modified, higher molecular weight forms of A20 (Supplementary Fig. 8A). A20 expression returned back to normal levels in cells restored with full-length HA-ATG16L1, but not when just the N-terminal domain was present (Supplementary Fig. 8A), suggesting that the WDD mediates the inhibitory effect that ATG16L1 has on A20 expression levels. Parallel NF-κB activation assays showed that Atg16l1-deficient MEFs respond better to TNF treatment (Supplementary Fig. 8B), pointing again to the idea that Atg16l1 represses NF-κB activation. Consistently, re-introduction of ATG16L1 or the N-terminal domain in Atg16l1$^{-/-}$ cells suppressed the signal, although the latter was less efficient (Supplementary Fig. 8B). These data argue that Atg16l1 regulates NF-κB signaling at least in part through a mechanism involving the WDD. Intriguingly, the NF-κB inhibitory potential of ATG16L1 in this context appears unrelated to its ability to regulate A20 expression, since Atg16l1-deficient cells actually show increased levels of A20 (see Supplementary Fig. 8A), and A20 is generally thought to repress the NF-κB pathway induced by TNF[13]. The notion that Atg16l1 regulates NF-κB independently of A20 is consistent with the results obtained in

reconstituted Atg16l1/A20-double deficient MEFs (see Supplementary Fig. 7B).

Together, these results point to an active post-transcriptional interplay between Atg16l1 and A20 in the regulation of each other's stability and ability to control autophagy and NF-κB activation, and suggest an important role of the WDD in the coordination of these activities.

**Spontaneous gut pathology in A20/Atg16l1 knockout mice**. To study the functional relationship between A20 and Atg16l1 in vivo, we next generated mice with double A20 and Atg16l1 deficiency in the intestinal epithelium. For this, A20 (A20$^{FL/FL}$) [12,14] and Atg16l1 floxed (Atg16l1$^{FL/FL}$) mice were crossed with Villin-Cre transgenic mice[41], generating IEC-specific A20/Atg16l1 double knockout mice (dKO) as well as wild-type (WT), A20$^{\Delta IEC}$ and Atg16l1$^{\Delta IEC}$ single knockout littermate controls (Fig. 5a). Double-deficient mice are born in normal numbers but are significantly smaller than their littermate controls (Fig. 5b–d). At the age of 5 weeks, A20/Atg16l1 dKO mice are on average 7 grams lighter than all 3 control groups of mice (Fig. 5d). In contrast to the 3 control groups, double deficient mice spontaneously develop colitis, as demonstrated by the presence of rectal prolapses (incidence of 40% in dKO, Fig. 5e) and increased vascularization and bowel wall thickening in the jejunum (Fig. 5f). High-resolution mini-endoscopy revealed a thickened and granular mucosal colon surface with altered vascularization, indicative of colitis in A20/Atg16l1 dKO mice. In contrast, WT, A20$^{\Delta IEC}$, and Atg16l1$^{\Delta IEC}$ display a normal mucosal surface with normal vascularization (Fig. 5g–i). Finally, increased expression of the inflammatory cytokines TNF and IL-1β and the chemokines MCP1 and CXCL2 could be detected in intestinal lysates from A20/Atg16l1 dKO mice but not in tissue lysates from littermate control mice (Fig. 5j), indicating spontaneous intestinal pathology only in mice with double A20 and ATG16L1 deficiency.

**Paneth/crypt cell defects in A20/Atg16l1 knockout mice**. In line with previous macroscopic observations, histological analysis of small intestine and colon tissue revealed severe jejunitis and colitis in A20/Atg16l1 dKO mice, characterized by a severely inflamed mucosa showing crypt elongation, villus blunting, presence of crypt abscesses, and immune cell infiltration (Fig. 6a–c). Alcian Blue-PAS (AB-PAS) stainings also show reduced numbers of goblet cells in sections of colon of dKO animals (Fig. 6d). In contrast, intestinal tissue of all three control groups

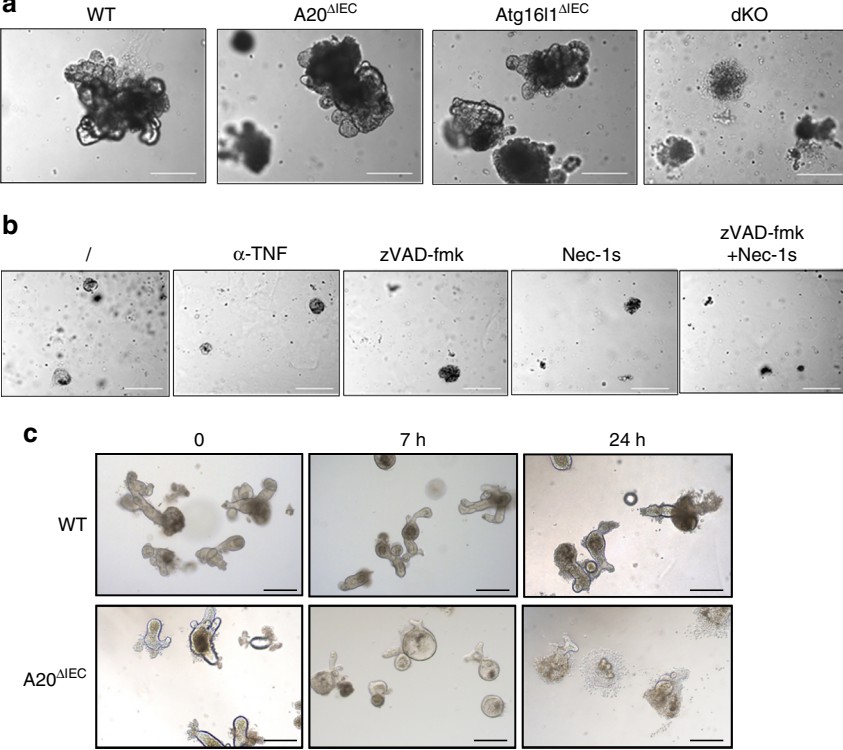

**Fig. 7** Combined A20/Atg16l1 deficiency induces IECs organoid lethality in vitro. **a** Organoids from A20/Atg16l1 dKO animals fail to grow in culture. Pictures of wild-type (WT), A20$^{\Delta IEC}$, Atg16l1$^{\Delta IEC}$, and dKO intestinal organoids, representative of three independent experiments. **b** Intestinal crypts from A20-Atg16l1 dKO mice cultured in the presence of anti-TNF, zVAD-fmk, Nec-1s, or zVAD-fmk + Nec-1s. **c** Wild-type (WT) and A20$^{\Delta IEC}$ organoids were incubated with supernatant from small intestinal explants from A20-Atg16l1 dKO mice for the indicated time points. Scale bars, 200 μm

(WT, A20$^{\Delta IEC}$, and Atg16l1$^{\Delta IEC}$) shows normal morphology (Fig. 6a–d). Histological scores could confirm small intestinal tissue inflammation and crypt loss in dKO tissue (Fig. 6e). This severe intestinal phenotype could already be observed in the small intestine and colon of very young, 3–4 weeks old, A20/Atg16l1 dKO mice (Supplementary Fig. 9).

Analysis of jejunal crypts of dKO mice by transmission electron microscopy (TEM) demonstrates Paneth cells with altered cell morphology displaying multiple pyknotic nuclear bodies indicating increased crypt cells apoptosis (Fig. 6f). Epithelial Paneth cell loss in dKO mice was confirmed by immunofluorescent staining of tissue sections for the Paneth cell antimicrobial protein Lysozyme P (Fig. 6g, h) and by quantitative PCR analysis on epithelial lysates for expression of the Paneth cell-specific genes cryptdin-1 (crypt-1) and lysozyme-P (LysP) (Fig. 6i). In agreement with a role for Paneth cells in producing trophic factors for intestinal stem cells[42], a significantly reduced expression of the intestinal stem cell marker Lgr5 could be demonstrated in small intestinal tissue of dKO mice (Fig. 6j). Despite showing no morphological Paneth cell defects, LysP and crypt1 expression in Atg16l1$^{\Delta IEC}$ is also reduced. Next, immunostaining for Ki67 revealed increased epithelial cell proliferation occupying the entire cell crypts including the stem cell region in the small intestine of dKO mice, in contrast to the small intestine of WT, A20$^{\Delta IEC}$ and Atg16l1$^{\Delta IEC}$ mice where active cycling Ki67-positive cells, so-called transit amplifying cells[43], are located above the stem cell compartment (Fig. 6k). In addition to the enhanced cell proliferation, histological analysis of the small intestine of dKO mice demonstrated increased apoptosis of crypt epithelial cells, as shown by TUNEL and cleaved caspase 3 staining, which is not observed in the intestinal tissue of the control groups (Fig. 6k–m). Although the pathology is most pronounced in the proximal small intestine jejunum,

similar but milder pathology was observed in the distal small intestinal tissue (Supplementary Fig. 10). Also colon tissue of 20-week-old dKO mice shows severe pathology with epithelial hyperproliferation and cell death in the distal parts of the colon (Supplementary Fig. 11), with only minor pathology in other parts (data not shown).

In conclusion, A20/Atg16l1 dKO mice display severe inflammation in both the small intestine and the colon, characterized by Paneth and goblet cell loss, IEC hyperproliferation and crypt cell apoptosis.

**A20/Atg16l1 deficiency prevents intestinal organoid culture.**
To investigate the interplay between A20 and Atg16l1 in a relevant in vitro model, and to determine whether the inflammatory phenotype in A20/Atg16l1 dKO mice results from an IEC intrinsic defect or from the inflammatory environment impacting on IEC homeostasis, we isolated small intestinal organoids from wild-type (WT), A20$^{\Delta IEC}$, Atg16l1$^{\Delta IEC}$, and A20/Atg16l1 dKO mice. However, and in contrast to the three other genotypes, we were not able to obtain organoid cultures isolated from dKO animals, and all dKO cells spontaneously start dying within one week of isolation. Moreover, during this short period of culture, dKO organoids display an abnormal phenotype and maintain a spheroid shape with minimal sproutings (Fig. 7a). Culturing A20/Atg16l1 dKO organoids with anti-TNF antibody, pan-caspase inhibitor zVAD-fmk or necroptosis inhibitor nec-1s could not increase the viability of dKO organoids (Fig. 7b). To further investigate the possible presence of cytotoxic factors in A20/Atg16l1 dKO mucosa, we incubated WT and A20 deficient organoid cultures with supernatant of ex vivo dKO small intestinal explants. Several hours post stimulation, the dKO explant supernatant induced marked swelling and subsequent death of

A20 deficient organoids while keeping WT control organoids alive (Fig. 7c). Multiple cytokines, including TNF, IL6, IFN-γ, CCL2 (MCP1), CXCL1 (KC), IL17, and IL22, were detected in dKO explant supernatant (data not shown), some of which (TNF and IFNγ) can induce cell death in A20 deficient enterocytes, as previously shown[12].

Together, these data indicate that the sensitivity of IECs to apoptosis causes the loss of intestinal barrier integrity and the development of severe inflammatory pathology in IEC-specific A20/Atg16l1 dKO mice.

## Discussion

Significant advances have been made in the understanding of IBD pathogenesis thanks to the identification of disease susceptibility genes in patients. Many of the IBD-associated genes identified so far are involved in the regulation of innate immune responses and intestinal epithelial functions. First evidence involving autophagy in IBD came with the identification of *ATG16L1*[5,15], findings which have shortly after been confirmed in experimental studies in mice with hypomorphic or deficient *Atg16l1* alleles[18,29]. However, the mechanisms by which autophagy controls intestinal homeostasis are still unclear. Autophagy is induced upon various types of cellular stress and contributes to innate immune responses. Hence, defects in autophagy may impair host defense and induce inflammatory reactions[16,44]. In this study, we investigated the importance of Atg16l1 for intestinal homeostasis by studying its interplay with the ubiquitin-editing protein A20, an essential negative regulator of inflammatory signaling, in IECs. Our results demonstrate that, in contrast to mice with single A20 or Atg16l1 deficiency in IECs that develop normally without intestinal defects, the combined loss of both proteins leads to spontaneous IBD-like pathology in mice. Substantial evidence has demonstrated cell death to be a central feature in IBD which can be triggered by epithelial cell death inducing barrier disruption, allowing infiltration of luminal bacteria into the mucosa[45]. Since both A20 deficient and Atg16l1 deficient IECs are highly sensitive to cytokine-induced cell death[12,14,31,32,46], this suggests that both proteins are indispensable to prevent barrier destruction in inflammatory conditions. In normal conditions with minimal cytokine exposure, A20 and ATG16L1 can compensate for each other's loss and preserve intestinal barrier integrity. However, upon loss of both A20 and ATG16L1, spontaneous barrier disintegration and chronic intestinal inflammation develops. These data support the growing notion that inflammatory signaling and autophagy cooperatively control intestinal homeostasis by preventing the death of enterocytes that would compromise intestinal barrier integrity[31,32,46]. Both apoptosis and necroptosis have been associated with IBD[47], and although apoptotic death is considered less inflammatory due to the containment of the cell content within apoptotic bodies, we believe that in our A20-Atg16l1 dKO model apoptosis of the intestinal epithelial cells is the driving cell death mode triggering chronic intestinal inflammation, given the clear observation of cleaved caspase-3 positive cells in dKO mice.

One of the phenotypes observed in A20/Atg16l1 dKO mice is a reduction in goblet and Paneth cells (particularly the latter), a phenomenon typically seen in IBD patients and in many experimental IBD mouse models[48]. Both are specialized secretory cell types responsible for the production of mucus and anti-microbial peptides, respectively, preventing bacterial infiltration, and hence are highly sensitive to ER stress and inflammatory cytokines[28-30,49]. A20 or Atg16l1 deficiency in IECs does not cause spontaneous Paneth or goblet cell death, but sensitizes to their loss in inflammatory conditions[12,14,31,32]. Our data showing that the combined deletion of A20 and Atg16l1 provokes loss of Paneth and goblet cells sustains the idea that defects in the innate defense mechanisms linked to these cells elicit barrier defects and contribute to the spontaneous development of intestinal pathology. These data are in line with previous work, showing that loss of either regulators of ER stress or modulators of autophagy result in each other's compensatory engagement and that spontaneous pathology only develops in case both mechanisms are defective[30].

We also identified a direct physical interaction and functional interplay between A20 and Atg16l1 that could constitute an important control point for intestinal homeostasis. First, we show that A20 and Atg16l1 physically interact through their OTU and WD40 domains, respectively. Second, we observed increased expression levels of Atg16l1 and LC3-II in A20-deficient conditions, reflecting enhanced autophagic flux promoted by a WDD-dependent unconventional activity of Atg16l1. Third, we found that A20 expression levels are induced in the absence of Atg16l1 due to the elimination of a WDD-mediated and lysosome-dependent activity of Atg16l1 that degrades A20. These data support the notion that Atg16l1 and A20 normally keep each other in check by mutually reducing their expression levels, a post-transcriptional cross-regulation that likely relies on their direct interaction (Supplementary Fig. 12A). Absence of one of the partners triggers a compensatory upregulation of the other that helps quench the aggressive intestinal inflammatory phenotype observed in double-deficient mice. The exact identity of the downstream pathways that mediate such compensation is unclear. Our data suggest that induction of Atg16l1 in A20-deficient cells promotes unconventional autophagy, enhanced p62 expression and decreased NF-κB activation that may help keep under control the elevated levels of NF-κB caused by absence of A20 (see summary in Supplementary Fig. 12B). Conversely, A20 upregulation in cells lacking Atg16l1 correlates with higher levels of p62, increased NF-κB activation and, according to the abundant literature on the functional role of A20[12-14], probably increased protection against cell death (Supplementary Fig. 12A). Therefore it seems that, instead of a single route that is coordinately regulated by Atg16l1 and A20, a complex mixture of signaling events involving NF-κB activation, induction of unconventional autophagy and (probably) protection from cell death are activated in single-deficient cells to compensate for the absence of the relevant partner. Unraveling exactly how these pathways become activated and interact with each other to deliver protective effects will require additional studies, but a few possible mechanistic hints are suggested by the literature.

For example, the observed increased expression levels of Atg16l1 in A20-deficient conditions suggest that prolonged NF-κB activation may play a role in Atg16l1 stabilization to induce upregulation of autophagy, as a compensatory mechanism[16]. In this context, IKKα activation, in response to cytokine and microbial stimulation, was recently shown to phosphorylate Atg16l1 leading to its stabilization, thus preventing ER stress[50]. A20 controls NOD2-induced NF-κB activation and induction of inflammatory signals through the deubiquitination of the adapter protein RIP2[51]. Although NOD2 can direct autophagy by recruiting Atg16l1 to the plasma membrane to impair bacterial entry[52], Atg16l1 was also shown to negatively regulate NOD2-dependent inflammatory responses independently of its canonical function in autophagy, by interfering with the ubiquitination and activation of RIP2[19]. This suggests that A20 might also be recruited to such complex and assist Atg16l1 in RIP2 deubiquitination, thus preventing chronic intestinal inflammation. However, these activities need further investigation.

The WDD is absent in yeast Atg16l1[20], suggesting that it mediates functions that are specific in multicellular organisms. Consistently, a deleted form of Atg16l1 lacking this region is fully competent to sustain basal and nutritional autophagy in

mammalian cells[21,22]. Recent evidence argues that the WDD acts as a docking site for upstream adapters able to engage the LC3-lipidation complex in unconventional compartments, thus promoting LC3-II synthesis through the N-terminal region that binds Atg5-Atg12[22,23,25,35]. Whether all activities carried out by the WDD involve LC3 lipidation or if other mechanisms may be involved is still unclear. However, irrespective of the underlying mechanism, this region appears to mediate some of the apparently non-autophagic functions where Atg16l1 has been implicated, like certain innate cellular responses against bacterial infection[52,53], trafficking of secretory vesicles in Paneth cells[29,54] or the control of inflammation[18,19]. Indeed, our results argue that the WDD does play a role in the regulation of the inflammatory response by interacting with relevant modulators of this pathway.

The WDD has been shown to interact with ubiquitin to capture invading pathogens and recruit them to the autophagic machinery[53]. In light of our results shown here, this process might be favored by A20, which also has ubiquitin binding activities[13]. A20 could also collaborate with Atg16l1 through p62. It is known that p62 can interact with TRAF6, facilitating its oligomerization and ubiquitination to activate downstream NF-κB signaling[55], implying that accumulation of p62 in A20 or Atg16l1-deficient conditions would enhance inflammatory stress. p62 also promotes caspase-8 aggregation leading to its full activation and apoptosis induction, a process which may be suppressed by A20[56,57]. Together, these studies suggest that p62 aggregates serve as signaling hubs that determine whether cells will survive through TRAF6-NF-κB signaling, or die through caspase-8 aggregation inducing cell death[55]. Both decisions might be regulated by A20.

In summary, our data reveal a close functional relationship between A20 and ATG16L1 to preserve intestinal barrier integrity and prevent intestinal inflammation. In addition, besides A20, our study also identified a number of other ATG16L1 WDD-interacting proteins regulating innate immunity and inflammatory signaling. More studies are needed to clarify the importance of these interactions for inflammatory signaling and tissue homeostasis.

## Methods

**Tissue-specific A20-Atg16l1-double deficient mice.** Conditional A20/*Tnfaip3* knockout mice, in which exons IV and V of *Tnfaip3* gene are flanked by two LoxP sites, were previously described[14]. Conditional Atg16l1 knockout mice, in which exon III of the *Atg16l1* gene is flanked by two LoxP sites, were generated using EUCOMM embryonic stem (ES) cells (ES cell clone EPD0102_2_A02). A20-floxed (A20[FL/FL]) and Atg16l1-floxed (Atg16l1[FL/FL]) mice were intercrossed and crossed with Villin-Cre transgenic mice[41], generating IEC-specific A20/Atg16l1 double knockout mice (A20[FL/FL]/Atg16l1[FL/FL]VillinCre[Tg/+] or dKO), as well as wild-type (WT), and single knockout A20 (A20[FL/FL]VillinCre[Tg/+] or A20[ΔIEC]) and Atg16l1 (Atg16l1[FL/FL]VillinCre[Tg/+], Atg16l1[ΔIEC]) knockout mice. Sex- and age-matched littermate mice were used in all studies and were co-housed to minimize microbiota influence. Experiments were performed on both male and female mice backcrossed into the C57BL/6 genetic background for at least eight generations. Mice were housed in individually ventilated cages in a specific pathogen-free facility. All experiments on mice were conducted according to institutional, national and European animal regulations. Animal protocols were approved by the ethics committee of Ghent University.

**Endoscopic analysis.** High-resolution mouse endoscopy and murine endoscopic index of colitis severity (MEICS) scoring was performed, as previously described[58], using a 'Coloview' endoscopic system (Karl Storz, Tuttlingen, Germany). Mice were anaesthetized with 2–2.5% isoflurane in oxygen during endoscopy.

**Isolation of intestinal crypts and 3-D organoid culture.** Intestinal organoids were derived from small intestine as previously described[59]. Briefly, a 10-cm piece of duodenum/jejunum was dissected and washed in phosphate-buffered saline (PBS). The intestine was opened longitudinally, villi were scraped away and the tissue was cut in 2–3 mm pieces. After thorough washing in PBS, pieces were incubated in 2 mM EDTA/PBS for 40 min at 4 °C on a rocking platform. After passages through a 70-μm cell strainers, four crypt fractions were isolated and purified by successive centrifugation steps. Four hundred microlitre of matrigel

(BD Biosciences) was added to crypt pellet and drops of 50 μl crypt-containing matrigel were added to pre-warmed wells of a 24-well plate and 20 μl to 8-well chamber (Bidi). After brief polymerization, 500 μl/200 μl of DMEM-F12-based complete growth medium, supplemented with N2 (Invitrogen), B27 (Gibco BRL), recombinant mEGF (Peprotech), R-Spondin1- and Noggin-conditioned media were added to each well and organoids were refreshed every 2 days. zVAD-fmk (50 μM) and Nec-1 s (1 mM) were diluted in the Matrigel before seeding and were added to the growth medium.

**Intestinal explants.** Small pieces of proximal small intestine (5 mm) were isolated, trimmed of fat and flushed with ice-cold sterile PBS/0.1% BSA. Intestinal pieces were weighed, cut longitudinally and placed in a 24-well plate. The explants were cultured over-night at 37 °C, 5% CO₂ in RPMI medium, supplemented with 10% FCS, penicillin, streptomycin, gentamycin, GlutaMAX™ (Gibco) and sodium pyruvate with intestinal lumen facing the medium. Supernatant of the explant culture was used to measure cytokine levels by Bioplex and results were normalized to tissue weight. After adding organoid growth factors, explant medium was used to stimulate organoid cultures.

**Proteomic studies.** JAR (human placenta choriocarcinoma) cells were transfected with the mammalian expression plasmid Peak12 driving the expression of GST, GST-WDD (320–607) or GST-WDD (231–607) open reading frames. THP-1 (human monocytic) cells were retrovirally transduced with the same constructs cloned into the P12-MMP retroviral vector[22]. For the definitive proteomics assay, a total of 15 × 6-cm plates of JAR cells were transfected per construct (GST or GST-WDD (231–607)) and lysed 48 h after transfection. THP-1 cells transduced with GST or GST-WDD (231–607) were divided into two groups. One of them was activated with LPS (Sigma; 5 μg/ml) and PMA (Sigma; 50 ng/ml) for 24 h to favor expression of inflammatory mediators whereas the other one remained untreated. 60 × 10 cm plates were processed per condition in this case since chimera expression was lower compared to JAR cells. Cells were lysed in a buffer containing 1% Igepal CA-630 (Sigma) for 30 min on ice with occasional mild vortexing. Lysates were cleared by centrifugation, diluted with the same buffer devoid of detergent to reach a final detergent concentration of 0.2%, and incubated for 3 h with agarose beads coupled to GSH (GE Healthcare) at 4 °C with rotation. Precipitates were washed twice with immunoprecipitation buffer (0.2% detergent), resuspended in 2x RSB and boiled. Samples were resolved in 10% poly-acrylamide gels and silver stained. Specific bands absent in control (GST) samples were excised and processed for proteomics identification. Protein digestion was performed as previously described[60] with minor variations. Silver-stained gel plugs were destained with a solution containing 7.5 mM potassium ferricyanide and 25 mM sodium thiosulfate, rinsed in water and dehydrated with 100% acetonitrile. Plugs were then DTT reduced and alkylated with iodoacetamide. Modified porcine trypsin (Promega, Madison, WI; 6 ng/μl in 20 mM ammonium bicarbonate) was added before incubation at 37 °C for 18 h. Tryptic peptides were dried in a speed vacuum system, desalted by using C18-homemade columns[61] and analyzed by reversed-phased LC-MS/MS using a nanoAcquity UPLC (Waters Corp., Milford, MA) coupled to a LTQ-Orbitrap Velos (Thermo-Fisher, San Jose, CA). Separations were done in a BEH 1.7 μm, 130 Å, 75 μm × 100 mm C18 column (Waters Corp., Milford, MA) at a 400-nL/min flow rate. Injected samples were trapped on a Symmetry, 5 μm particle size, 180 μm × 20 mm C18 column (Waters Corp., Milford, MA). Peptides were eluted using a 30 min gradient from 3 to 35% B (0.5% formic acid in acetonitrile). The LTQ-Orbitrap Velos was operated in a data-dependent MS/MS mode using Xcalibur (Thermo-Fisher, San Jose, CA). Survey scans were acquired in the mass range 400–1600 m/z, with 30,000 resolution at *m/z* 400 and lock mass option enabled for the 445.120025 ion[62]. The 20 most intense peaks having ≥2 charge state and above 500 intensity threshold were selected in the ion trap for fragmentation by collision-induced dissociation. MASCOT [v 2.3] and Sequest HT [v 1.3] search algorithms were used for searching the acquired MS/MS spectra using Thermo Scientific Proteome Discoverer software (v. 1.4.1.14) against a database of human sequences (Uniprot, release 2013_05) with common contaminants. Search parameters were as follows: fully tryptic digestion with up to two missed cleavages, 10 ppm and 0.8 Da mass tolerances for precursor and product ions, respectively, oxidation of methionine was established as variable modification and carbamidomethylation of cysteine as fixed modification. 1% false discovery rate using Percolator was used for peptide validation. The crude lists of identified proteins (Supplementary Data 1) were manually curated to eliminate non-human references and redundant hits, and to find in all cases the equivalent reviewed Uniprot reference. We implemented a filtration step using the Crapome sever (http://crapome.org/) to eliminate hits present in more than 10% of the control proteomics assays included in this database (thus likely to be artifacts). Such curation steps resulted in secondary lists of non-redundant and fully reviewed Uniprot entries annotated for function (Supplementary Data 2). The reviewed lists were subjected to a Venn analysis to find out the degree of redundancy of the identified proteins in the three experimental systems analyzed (Supplementary Data 3). Proteins were further classified according to their involvement in different functions/biological activities as annotated in any of the two Uniprot fields: Function [CC] and/or Gene Ontoloty (Biological process). Selected candidates are shown in Supplementary Data 4. The mass spectrometry proteomics data have been deposited to the ProteomeXchange Consortium via the PRIDE partner

repository with the dataset identifier PXD013180 (https://doi.org/10.6019/PXD013180).

**DNA constructs**. HA-NALP2 and Flag-MDA5 mammalian expression plasmids were kindly provided by Dr. Fernández-Luna (Hospital Marqués de Valdecilla, Santander, Spain) and Dr. Reis e Sousa (Cancer Research Institute, London, UK), respectively. ATG16L1 constructs were previously described[22]. A20 deletions were generated by PCR using the oligonucleotides shown in Supplementary Table 1, and subcloned in frame into a pCDNA3-HA plasmid (Not1 to Xho1). Mutations were introduced by site-directed mutagenesis using oligonucleotides shown in Supplementary Table 2. All constructs were verified by sequencing.

**Transfections and retroviral transductions**. Transfections were carried out using the JelPEI transfection kit following manufacturer's instructions. VSV-G pseudo-typed retroviral particles were produced in HEK-293T cells by cotransfecting the relevant constructs cloned into the P12-MMP retroviral vector with the helper plasmids pMD.gag-pol and pMD-G (expressing gag-pol and VSV-G env, respectively). The retroviral NF-κB-luciferase reporter vector was obtained from Dr. Felix Randow (LMB, Cambridge, UK). Infections were done by diluting the viral supernatants 1:1 with fresh medium and centrifuging the resulting mix onto the target cells in the presence of polybrene (8 μg/ml) for 1 h at 2000 rpm/32 °C.

**Luciferase activity assays**. Cells harboring a retroviral NF-κB-luciferase reporter system were plated in 24-well plates in low serum conditions (0.5% FCS) and next day treated with TNF. Cells were then lysed and processed for luciferase measurements using a luciferase assay kit (Promega) following the instructions provided by the manufacturer.

**Co-immunoprecipitation studies**. The 6-cm plates of HEK293T cells were transfected with a total of 10 μg plasmid mix. The cells were lysed in 200 μl of 1% NP-40 lysis buffer for 30 min on ice and spun down top speed at 4 °C for 10 min to remove the nuclei. Supernatants were diluted with the same lysis buffer without the detergent to reach a 0.2% final concentration of NP-40 to preserve protein-protein interactions. Protein lysates were then pre-cleared with 20 μl agarose-proteinG for 1 h at 4 °C on rotating wheel. IP was performed with 20 μl agarose-GST for 3 h at 4 °C on rotating wheel or with primary antibodies (control Ig: rabbit anti-AU1 Covance PRB-130P; anti-ATG16L1: rabbit polyclonal MBL PM040 1:150) for 2 h plus 1 h with agarose-protein G beads at 4 °C on rotating wheel. Samples were washed with cold 0.2% NP-40 lysis buffer, resuspended in 25 μl 2xRSB, boiled, spun down and subjected to Western blotting using PVDF membranes (Millipore).

**MEF isolation**. 15.5 dpc embryos from either A20[FL/FL] were isolated and mouse embryonic fibroblasts (MEFs) were prepared. Cells were immortalized through serial passaging and backups stored in liquid nitrogen.

**Western blot**. MEF, organoids and IEC samples were lysed in E1A buffer supplemented with sodium orthovanadate, sodium fluoride and cOmplete™, EDTA-free Protease Inhibitor Cocktail (Roche) for 5–10 minutes on ice, spun down cold at maximum speed and protein concentration was measured with Bradford. 20–35 μg of protein was loaded on the gels and separated by SDS-PAGE (PAGE), transferred to nitrocellulose or PVDF (Millipore) membranes, and analyzed by immunoblotting. Proteins were detected with following antibodies: mouse anti-A20 (Santa Cruz sc-166692, 1:1000), rabbit anti-Atg16l1 (Cell Signaling 8089, 1:1000); rabbit anti-LC3 (MBL PM036, 1:1000); rabbit anti-p62 (MBL PM045, 1:2000), goat anti-IkBa (Santa Cruz sc-371-G, 1:1000); mouse anti-P-IkBa (Cell Signaling 9246, 1:1000); mouse anti-actin (MP Biomedicals 8691002, 1:10,000); mouse anti-HA (BioLegend 901501, 1:1000); mouse anti-Flag (Sigma F1804, 1:1000); rabbit anti-GST (Cell Signaling 2622, 1:1000); mouse anti-Tubulin (Sigma T4026, 1:40,000); mouse anti-GAPDH (Abcam Ab8245, 1:10,000); mouse anti-Atg16l1 (MBL 150-3, 1:2000); mouse anti-LC3 (MBL M186-3, 1:2000); mouse anti-UB (FK2; Millipore 4-263, 1:1000). As secondary antibodies, anti–rabbit, anti-mouse (GE Healthcare) or anti-goat (Santa Cruz)-HRP conjugates were used, and the signal was detected with enhanced chemiluminescence substrate ECL (Perkin Elmer).

**Quantitative real-time PCR**. For RNA extraction, two 8 cm-long representative segments of small intestine and one piece of colon were flushed with PBS to remove fecal matter. One end was ligated and pieces were filled with Total RNA lysis buffer (Bio-rad) supplemented with β-mercaptoethanol, lysed on ice for 5 min and the lysates were then snap frozen in liquid nitrogen. For RNA isolation, Aurum Total RNA mini kit (Bio-rad) was used. RNA was reverse-transcribed with iScript Advanced kit (Bio-rad) and qPCR was performed with SybrGreen mix (Roche) on a Light Cycler 480 (Roche) in triplicates. The following mouse-specific primers were used: TNF forward ACCCTGGTATGAGCCCATATAC; TNF reverse ACACCCATTCCCTTCACAGAG; IL1β forward CACCTCACAAGCAGAGCACAAG; IL1β reverse GCATTAGAAACAGTCCAGCCCATAC; CXCL2 forward ACAGAAGTCATAGCCACTCTC, CXCL2 reverse TTAGCCTTGCCTTTGTTCAG; MCP1 forward GCATCTGCCCTAAGGTCTTCA, MCP1 reverse TGCTTGAGGTGGTTGTGGAA; Lysozyme-P forward GCCAAGGTCTAACA

ATCGTTGTGAGTTG, Lysozyme-P reverse CAGTCAGCCAGCTTGACACCACG; Cryptdin-1 forward TCAAGAGGCTGCAAAGGAAGAGAAC, Cryptdin-1 reverse TGGTCTCCATGTTCAGCGACAGC; Lgr5 forward AGAACACTGACTTTGAATGG, Lgr5 reverse GACAAATCTAGCACTTGGAG. Data were analyzed on Light Cycler Software and qBase + (Biogazelle) and normalized to three reference targets Eef1a1, Matr3, and Cox4i1 (Cox4i1 forward AGAATGTTGGCTTCCAGAGC; Cox4i1 reverse TTCACAACACTCCCATGTGC; Eef1a1 forward TCGCCTTGGACGTTCTTTT; Eef1a1 reverse GTGGACTTGCCGGAATCTAC; Matr3 forward TGGACCAAGAGGAAATCTGG; Matr3 reverse TGAACAACTCGGCTGGTTTC). For analysis of Atg16l1 expression in MEFs and organoids, qPCR was performed using an Atg16l1-specific TaqMan probe (Mm00513085_m1, Thermo Fisher) on a Light Cycler 480 (Roche) in triplicates. Data were analyzed in qBase + (Biogazelle) and normalized to two reference targets for MEFs, Tbp, and B2m, and to three reference targets for organoids GAPDH, Tbp, and B2m (GAPDH Mm99999915_g1; Tbp Mm00446971_m1; B2m Mm00437762_m1; Thermo Fisher).

**Tissue sample preparation**. Freshly isolated small intestinal segments (duodenum, jejunum and ileum) and colon were flushed with PBS to remove the fecal matter and subsequently flushed with 10% formalin (Sigma Aldrich) and fixed at room temperature on orbital shaker. Formalin was removed and intestines were put in 70% ethanol, then processed and dehydrated before embedding in paraffin wax using standard automated methods.

**Histology and immunofluorescence**. Intestinal organoids were mechanically released from matrigel using cold PBS, fixed in 4% PFA for 1 h at room temperature and permeabilized. Formalin-fixed tissue was embedded in paraffin and 5 μm sections were cut and stained with haematoxylin/eosin. For combined AB and PAS stainings, dewaxed sections were hydrated and incubated in Alcian Blue for 20 min. Sections were then washed with water before incubation in 1% periodic acid for 10 min and followed by incubation in Schiff's reagent for 10 min. Tissues were counterstained with Mayer's haematoxylin for 30 s, washed and dehydrated before mounting with Depex. For immunochemistry, sections were dewaxed and incubated in Vector antigen unmasking solution antigen retrieval solution and boiled for 20 min in a Pick cell cooking unit and cooled down for 2.5 h. Endogenous peroxidase activity was blocked with 3% peroxidase-blocking buffer (Sigma) for 30 min at room temperature. Blocking buffer (fish skin gelatin with 5% normal goat serum) was added to the slides for 1 h at room temperature. Primary antibodies (rabbit anti-Ki67, dilution 1/1000, Cell Signaling 12202; rabbit anti-lysozyme, dilution 1/700, Dako A0099; anti-cleaved caspase-3, 1/700 dilution, Cell Signaling 9661) were incubated overnight in blocking buffer at 4 °C. Slides were then incubated with biotinylated secondary antibodies for 1 h at room temperature and then incubated with avidin-biotin complexes (AB, Vector Labs) and peroxidase activity was detected with diaminobutyric acid (DAB) substrate (Vector Labs). Slides were counterstained with Mayer's haematoxylin and mounted in Depex mounting medium. For lysozyme staining, sections were incubated with DyLight-488 conjugated goat anti-rabbit secondary antibody (1:500 dilution, Fisher Bioblock Scientific), and cell nuclei were counterstained with DAPI (40, 6-diamidino-2-phenylindole, Invitrogen) in ProLong Gold anti-fade reagent. Slides were mounted with Entellan. Apoptosis was analyzed with an in situ cell death detection kit (TMR-red, Roche). For double-staining on organoids, TUNEL was performed after permeabilization, followed by the blocking and incubation with primary antibody. Endogenous GFP signal was imaged on Leica SP5 confocal microscope after DAPI-counterstain. Fluorescence imaging was done at an SP5 confocal microscope (Leica). Confocal images were represented as maximum projections.

**Histological scoring**. To quantify the degree of pathology, representative sections of small intestine were scored blindly using the modified scoring scheme from ref. [63]. Paneth cells were quantified based on the selected region of interest in Fiji ImageJ software, relative to the area selected.

**Quantification of cleaved caspase-3[+] and TUNEL[+] cells**. Jejunal and colonic sections of three different sections were analyzed per mouse and 10 consecutive crypts within each section were counted for presence of cleaved-caspase-3 or TUNEL-positive cells. Averages per crypt are displayed.

**Transmission electron microscopy**. Small intestinal tissue was cut into pieces and immersed in a fixative solution of 2.5% glutaraldehyde, 4% formaldehyde in 0.1 M sodium cacodylate buffer, placed in a vacuum oven for 30 min and then incubated for 3 h at room temperature with gentle rotation. After adding fresh fixative, the tissue was incubated overnight at 4 °C, washed three times for 20 min with buffer solution and post-fixed in 1% $OsO_4$ with $K_3Fe(CN)_6$ in 0.1 M sodium cacodylate buffer, pH 7.2 at room temperature for 1 hour. After washing in $ddH_2O$, samples were dehydrated through a graded ethanol series, including a bulk staining with 2% uranyl acetate at the 50% ethanol step followed by embedding in Spurr's resin. Semi-thin sections were cut at 0.5 μm and stained with toluidine blue. Ultrathin sections of a gold interference color were cut on an ultra-microtome (Leica EM UC6), followed by post-staining with uranyl acetate and lead citrate in a Leica EMAC20 and collected on formvar-coated copper slot grids. Sections were imaged

on a JEM 1400-plus transmission electron microscope (JEOL, Tokyo, Japan) operating at 60 kV.

**Statistical analysis**. Results are expressed as the mean±SEM. Body weight, qPCR data, MEICS and histological score data were analyzed using one-way ANOVA and Kruskal-Wallis nonparametric test, corrected for multiple comparisons. Data were analyzed using GraphPad Prism 7 software. Body weight data over time were analyzed as repeated measurements using the residual maximum likelihood as implemented in Genstat v17. The following linear mixed model (random terms underlined) was fitted to the repeated weight loss data: the weight loss calculated for the ith mice from genotype $j$ ($j = 1… 2$; control and knockout) measured at day $t$ ($t = 1… 18$), and where $\mu$ is the overall mean of weight loss calculated for all mice across all time points. The random experiment effects in the model were assumed to be independent and normally distributed. Times of measurement were set as equally spaced. The correlation structure was modeled as autoregressive order 1 (AR1), allowing heterogeneity over time, and was selected as best model fit based on a likelihood ratio test statistic and the Aikake information coefficient. Significance of the fixed main and interaction effects was assessed by an $F$-test.

**Reporting summary**. Further information on experimental design is available in the Nature Research Reporting Summary linked to this article.

## Data Availability

The mass spectrometry proteomics data have been deposited to the ProteomeXchange Consortium via the PRIDE partner repository with the dataset identifier PXD013180 (https://doi.org/10.6019/PXD013180). Raw data associated with all reported averages in graphs and charts, as well as uncropped versions of all blots presented in the Figures and the Supplementary Figures are provided as Source Data File. All other data are available from the authors upon reasonable requests.

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

## Acknowledgements

We thank Dr. Felix Randow (LMB, Cambridge, UK) for sharing the retroviral NF-κB-luciferase reporter vector. K.S. was a predoctoral fellow with the Institute for the Promotion of Innovation by Science and Technology (IWT) and was supported by a "Kom op tegen Kanker" (Stand up to Cancer) grant from the Flemish cancer society. We thank the VIB Bioimaging Core for training and access to the instrument park. We thank Femke Baeke and Michiel De Bruyne for EM analysis, Amanda Goncalves for imaging support, Benjamin Pavie for lysosomal quantification, Marnik Vuylsteke for statistical analysis and Laetitia Bellen for animal care. Work in the G.v.L. lab is supported by research grants from the FWO, the "Geneeskundige Stichting Koningin Elisabeth" (GSKE), the CBC Banque Prize, the Charcot Foundation, the "Belgian Foundation against Cancer", and the "Concerted Research Actions" (GOA) of the Ghent University. I.S.G. and E.B.R. were supported by Fundación Moraza (USAL) and FPU (Spanish Government) predoctoral fellowships, respectively. F.X.P. was funded by grants SAF2014-53320-R and SAF2017-88390-R from the Spanish Government, IBD-0369 from the Broad Medical Research Program (Crohn's and Colitis Foundation, USA) and FIC016U14 from the Junta de Castilla y León local government. The CIC proteomics service was funded by grant PRB3 (IPT17/0019—ISCIII-SGEFI/ERDF).

## Author contributions

K.S., L.V., F.X.P., and G.v.L. designed the study; K.S., I.S.G., E.B.R., A.M., M.S., I.P. H.K.V., and R.D.R. performed the experiments; K.S., E.P., S.L., S.S., A.W., L.V., F.X.P., and G.v.L analyzed the data; K.S., L.V., F.X.P., and G.v.L. wrote the manuscript.

## Additional information

**Competing interests:** The authors declare no competing interests.

