## [Peer Review File · Nature Communications]

Reviewers' comments:

Reviewer #1 (IBD, gut barrier)(Remarks to the Author):

Slowicka et al present an interesting study with a novel finding that has biological significance and clinical relevance for IBD. Both A20 and ATG16L1 have been linked mechanistically and genetically with IBD and this manuscript describes a biochemical association between these molecules, and a phenotype of colitis when both are knocked out in intestinal epithelium. The experiments are generally well performed and the manuscript is well written. The weaknesses include remaining uncertainties around both the biochemical association/cellular mechanisms and the functional consequences of the deficiency. However, fully addressing these constitutes a large amount of work that is opened up by the interesting primary findings presented here. The paper would be strengthened by addressing the specific recommendations below.

Major Points

1. A biochemical association between A20 and ATG16L1 WDD domain is clearly shown with well performed and presented IP experiments and Western blots with appropriate controls. A20 contains 7 repeats of a motif known to bind the the ATG16L1 WDD domain and each of these was mutated without an effect of binding. It would be better to together mutate each of these domains as it quite likely that there is redundancy or several domains required for binding.
2. The effect of A20 on Atg16l1 protein expression in MEFs and intestinal organoids is interesting but not fully explored. Experiments should be conducted to explore whether this is regulated at the mRNA and/or protein level as this is critical to a potential role of the A20/Atg16l1 association. Furthermore, the organoid result in Fig 4B is not convincing and both 4A and 4B results should be accompanied by densitometric analyses showing the reproducibility of this finding. The data not shown on lack of ubiquitination of Atg16l1 and lack of stabilisation on proteasome inhibition should be included at least as Supplementary data because, although negative, these are important findings.
3. The phenotype in the IEC DKO mice is marked, but could be described in more detail. While the main emphasis is that there is enhanced apoptosis, in fact in vivo there is marked proliferation and even appears signs of dysplasia potentially in the SI. The histological phenotype requires more detailed description and addition of scoring of inflammation, crypt elongation etc across all regions of the small and large intestine. Clarity surrounding whether the primary defect is in secretory cells or affecting stem cells, enterocytes etc would aid interpretation. The lack of growth of the organoids from DKO mice is interesting but not fully explored – while anti-TNF did not work it would be good to explore chemokine release by the DKO epithelium and in fact how organoids would respond to the inflammatory environment established in the mice in vivo where hyperproliferation is the phenotype.

Minor Points

1. Figure legend 3 refers to IP methods in Fig 1 – this should be Fig 2.

Reviewer #2 (Autophagy, IBD)(Remarks to the Author):

The authors report a protective role for A20 and ATG16L1 in intestinal epithelial cells. They expressed GST1 ATG16L1-231-607 in JAR (choriocarcinoma) and THP-1 (monocytic) cells for proteomic studies at baseline and after LPS stimulation to identify interaction partners of the WDD domain. The authors found ~1000 interacting proteins and specifically NALP2, NALP4 and A20 which were confirmed to interact with Atg16l1 WDD by co-ip experiments in HEK293T cells. The OTU domain (with deubiq. activity) in A20 is required for the interaction with ATG16L1. The T300A ATG16L1 Crohn's disease variant does not impact on the interaction with A20 and also IBD risk conferring A20 variants in the

OTU domain did not impact on ATG16L1 binding. A20 knockout MEFs exhibited increased NF- κ B activation (as previously reported) which associated with increased ATG16L1 expression and LC3 conversion indicative for autophagy induction at baseline. Autophagy induction was not pronounced after TNF stimulation. The authors generated A20/Atg16l1-IEC specific double KO mice which exhibit spontaneous small and large intestinal inflammation. Intestinal inflammation associated with an altered morphology of goblet and Paneth cells (i.e. defects in antimicrobial expression and possibly death) and caspase 3+ death of crypt-based IECs. Organoids from DKO mice were not viable and could not be rescued by pharmacologic inhibition of apoptosis or necroptosis.

Identification of interaction partners of ATG16L1 is of interest to the field. The authors demonstrate that A20 (besides a plethora of other proteins) and ATG16L1 physically interact in cells other than intestinal epithelium. The mechanism how A20 regulates ATG16L1 level is unclear but may be explained by work from Diamanti et al (JEM 2017). When the physical interaction between A20 and ATG16L1 fails due to deletion of both proteins in intestinal epithelial cells, spontaneous ileo-colitis develops which is associated with IEC death. These data suggest that A20 and ATG16L1 are critical to maintain IEC homeostasis. Whether cell death is a consequence of an interplay between A20 and ATG16L1 is unclear because the relevance of a direct physical interaction between A20 and ATG16L1 is unresolved. It appears that the presented protein studies are uncoupled from the notable phenotype they observe in DKO mice. Relevant questions that would address a potential interplay between A20 and ATG16L1 have not been addressed in vitro or in vivo. For example, is A20 required for autophagy in IECs or do the authors observe induction of autophagy in IECs from A20-ko mice? Does ATG16L1-mediated autophagy degrade A20 to control NF- κ B signalling? Which signalling event elicits intestinal inflammation? What is the mode of cell death? Is death cause or consequence of disease?

Further general remarks:

The authors utilise non-epithelial cell lines for their proteomics analysis, the interactome in IECs may be different.

Figure 4: LC3 conversion (indicative for autophagy induction) occurs at baseline but not after TNF stimulation (A). This finding may be important and requires proper immunoblots without blurred LC3 bands (bubbles?) and irregular B-Actin. In contrast A20-/- small intestinal epithelial organoids did not exhibit increased Atg16l1 expression or LC3 conversion at baseline. As such, the regulation in MEFs and IEC organoids appears different as it stands now. If the authors claim induction of autophagy in A20-/- cells, they must use more assays than an immunoblot of LC3 to proof this concept.

Figure 5A: Is Atg16l1 required for A20 expression or does it stabilise A20? Is it transcriptional Regulation of Atg16l1 by A20?

Does a-tocopherol or ferrostatin-1 protect against cell death of A20/Atg16l1-/- organoids?

As it stands now, the title is overstated as it implies a direct link between a common pathway controlled by A20 and ATG16L1 that instigates intestinal inflammation. It may be that the mere lack of both proteins induces death of IECs due to important homeostatic (independent) functions of both proteins.

Discussion:

1. The notion that "Atg16L1 deficiency in IECs does not cause spontaneous Paneth cell defects" is wrong. (see work from Cadwell, Adolph and Conwell et al.)
2. "Our data showing that the combined deletion of A20 and Atg16L1 provokes loss of Paneth and goblet cells sustains the idea that defects in the innate defense mechanisms linked to these cells

contribute to the spontaneous development of intestinal pathology. These data are in line with previous work, showing that loss of either regulators of ER stress or modulators of autophagy result in each other's compensatory engagement and that spontaneous pathology only develops in case both mechanisms are defective (Adolph et al 2013).” The referenced study in this context does not support the notion that dying IECs such as Paneth cells trigger intestinal inflammation. This study demonstrated that stressed Paneth cells (that do not die) elicit intestinal inflammation. Other studies indeed demonstrated that dying IECs (and the absence of Paneth or goblet cells) elicit a barrier defect and consequently intestinal inflammation (Gunter C Nature 2008).

Point-by-point response to reviewer's comments

We would like to sincerely thank the reviewers for their instructive and considerate comments. We have put in a considerable effort to meet all of the points that were raised by conducting a series of novel experiments and by adapting the manuscript text, as appropriate. Detailed responses follow:

Reviewer #1

Slowicka et al present an interesting study with a novel finding that has biological significance and clinical relevance for IBD. Both A20 and ATG16L1 have been linked mechanistically and genetically with IBD and this manuscript describes a biochemical association between these molecules, and a phenotype of colitis when both are knocked out in intestinal epithelium. The experiments are generally well performed and the manuscript is well written.

We greatly appreciate the referee's comment on the novelty and significance of our study.

The weaknesses include remaining uncertainties around both the biochemical association/cellular mechanisms and the functional consequences of the deficiency.

We appreciate the reviewer's insightful comments and suggestions for improvement of our study. In the following we provide a point-by-point reply to the individual comments.

Major Points

1. A biochemical association between A20 and ATG16L1 WDD domain is clearly shown with well performed and presented IP experiments and Western blots with appropriate controls. A20 contains 7 repeats of a motif known to bind the ATG16L1 WDD domain and each of these was mutated without an effect of binding. It would be better to together mutate each of these domains as it quite likely that there is redundancy or several domains required for binding.

We agree. As suggested by the reviewer we have simultaneously mutated all 7 WDD-binding motifs present in the N-terminal fragment of A20 that suffices to interact with the WDD (A20-92-263). Co-immunoprecipitation studies showed that the mutant A20-92-263 variant (7M) is unable to interact with GST-WDD, suggesting that the different motifs cooperatively participate in interacting with the WDD. This new information is now included in Supplementary Fig. 3C of the revised manuscript. New text describing this figure has been inserted in page 7.

2a. The effect of A20 on Atg16l1 protein expression in MEFs and intestinal organoids is interesting but not fully explored. Experiments should be conducted to explore whether this is regulated at the mRNA and/or protein level as this is critical to a potential role of the A20/Atg16l1 association.

We agree and have now measured mRNA levels of Atg16L1 in WT and A20 deficient MEFs and organoids. However, no significant difference in Atg16L1 mRNA expression can be measured between both genotypes, concluding that the effect of A20 on Atg16L1 expression is regulated at the protein level. These data have now been included in the revised version of the manuscript (Figure 4C and D).

We have conducted a number of experiments to try to establish the relevance of A20/Atg16L1 interaction in the context of our study. The new results are discussed below and, although they do not directly address the comments of the reviewer in this point, we think they are relevant to the general issue of what is the "potential role of the A20/Atg16L1 association", as stated above by the reviewer.

To address some of the concerns raised by the reviewer, and also to fulfil the comments provided by reviewer 2 (see below), we reconstituted Atg16L1/A20-double deficient

MEFs with retroviral constructs expressing HA-A20 and/or different versions of ATG16L1 (full-length or just the N-terminal domain (delta-WDD)). Initial experiments were directed to recapitulate the previously observed stabilization of endogenous Atg16L1 in the absence of A20 (Fig. 4A), in order to subsequently explore the requirement of the WDD in this process. However, such efforts were unsuccessful, perhaps because the subtle effect observed for endogenous Atg16L1 is lost when both A20 and ATG16L1 are expressed at higher levels by retroviral transduction.

However and unexpectedly, in these experiments we found that ATG16L1 causes destabilization of A20, since MEFs expressing transduced ATG16L1 showed reduced levels of A20 (particularly after treatment with TNF; see the new Figs. 4E and 4F). Such destabilization is apparently mediated by the WDD, because absence of this region recovered A20 expression (Fig. 4E and 4F). Importantly, to generate this cellular system, cells were transduced first with HA-A20 and later with the different ATG16L1 constructs, so any change in A20 expression between the different strains cannot be caused by a dissimilar transduction efficiency of A20. Considering that A20 is expressed from a constitutive retroviral construct, the observed destabilization of A20 is likely a post-transcriptional event. In fact, treatment with lysosomal inhibitors (baflomycin, E64d/pepstatin) tended to normalize A20 expression in cells harbouring different ATG16L1 status (no Atg16L1, full-length ATG16L1 or just the N-terminal domain; see the new Supplementary Fig. 7A), arguing that a lysosomal degradation pathway (likely autophagy) functions more intensely in the presence of full-length ATG16L1 to eliminate A20. In addition, we provide new data indicating that endogenous A20 is also regulated by ATG16L1 through the WDD. Thus, Atg16L1-deficient cells treated with TNF showed increased levels of A20 associated with the appearance of higher molecular weight forms of this protein (new Supplementary Fig. 8A). Such increased expression of A20 is returned back to normal levels upon re-expression of full-length ATG16L1 but not in cells restored with just the N-terminal domain (Supplementary Fig. 8A), suggesting again that the WDD is important for A20 destabilization by ATG16L1. Since the WDD mediates the direct interaction between A20 and ATG16L1, we believe that the A20 destabilizing activity of ATG16L1 likely requires direct binding between both molecules.

Given the important role of A20 in the control of NF- κ B activation induced by TNF, we measured the possible impact of the varying expression levels of A20 in the NF- κ B response using a suitable NF- κ B-luciferase reporter system. While we were unable to evaluate this issue in cells restored with HA-A20 (since overexpression of A20 dominantly inhibited NF- κ B activation in all strains; see the new Supplementary Fig. 7B), we found that, in the absence of A20, Atg16L1 reduces NF- κ B activation, because A20/Atg16L1-double deficient MEFs restored with ATG16L1 responded less efficiently to TNF treatment (Supplementary Fig. 7B). Again, the WDD seems to be important in this context, because cells expressing just the N-terminal domain of ATG16L1 behaved as Atg16L1-deficient cells (Supplementary Fig. 7B).

Similar studies in Atg16L1-deficient MEFs showed that absence of Atg16L1 increases NF- κ B responsiveness to TNF (new Supplementary Fig. 8B), suggesting again that Atg16L1 represses NF- κ B activation. Consistently, re-expression of ATG16L1 or just the N-terminal domain suppressed the signal, although the effect of the latter was less pronounced (Supplementary Fig. 8B). These data argue again that the WDD mediates at least in part the suppressive effects of Atg16L1 in this context. Interestingly, given that Atg16L1-deficient cells show increased A20 expression (Supplementary Fig. 8A) and considering that A20 is generally thought to repress NF- κ B activation in the TNF pathway, the NF- κ B inhibitory function of Atg16L1 is probably unrelated to its ability to control A20 expression levels. This notion is also sustained by the results in Supplementary Fig. 7B, where we show that transduced ATG16L1 inhibits NF- κ B activation by TNF in A20-deficient cells. Taken together, these results suggest that Atg16L1 regulates TNF-induced NF- κ B activation at least in part through its WDD and

independently of A20.

We conducted a series of additional studies to explore the nature of the increased autophagic response observed in A20-deficient MEFs (as shown in Fig. 4A). In these studies we found that accumulation of LC3-II likely reflects a more active autophagic flux in this system, since A20^{-/-} cells showed increased LC3-II levels even in the presence of the lysosomal inhibitor bafilomycin (see the new Supplementary Fig. 5A for basal autophagy, and Supplementary Fig. 5B for TNF-induced autophagy). However, the idea that A20-deficient cells have a hyperactive autophagic pathway is contradicted by data indicating a parallel accumulation of p62 (see Fig. 4A). Since p62 is usually degraded upon effective autophagy, its stabilization is normally interpreted a symptom of blunted autophagic flux. Interestingly, expression of the WDD in A20^{-/-} cells (via retroviral transduction) inhibited in a dominant-negative manner the accumulation of LC3-II induced by TNF (new Supplementary Fig. 5C), indicating that the autophagic response detected in these conditions has unconventional features (WDD-mediated) that could help explain the apparently contradicting behavior of LC3-II and p62 in this setting. Alternatively, p62 is an early NF- κ B response gene (Vadlamudi and Shin, *Febs Lett.*, 1998), and therefore likely to be overexpressed in A20 conditions.

Lastly, we provide new results showing induced interaction (co-precipitation) between endogenous A20 and transduced ATG16L1 in response to TNF both in MEFs and HCT116 cells (see the new Fig. 3C). In our view, these data help strengthen the notion that both molecules functionally interact through their physical association in response to physiological stimuli.

Therefore, taken together, all these results point to the idea that there is a tight interplay between A20 and Atg16L1 where their direct interaction through the WDD controls each other's stability (see the new Supplementary Fig. 12A), thus providing a possible mechanistic basis for their genetic interaction in the control of intestinal homeostasis. Loss of one interaction partner results in a compensatory upregulation of the other, resulting in only sensitization but no spontaneous epithelial cell death (cfr. mild phenotype in single-deficient mice). In contrast, losing both interaction partners results in spontaneous enterocyte cell death and intestinal inflammation. Our results in MEFs show that stabilization of Atg16L1 in A20-deficient cells promotes unconventional (WDD-dependent) autophagy, enhanced p62 expression and decreased NF- κ B activation that may help keep under control the elevated levels of NF- κ B caused by absence of A20 (see summary in the new Supplementary Fig. 12B). Conversely, A20 upregulation in cells lacking Atg16L1 correlates with higher levels of p62, increased NF- κ B activation and, according to the abundant literature on the functional role of A20, probably increased protection against cell death (Supplementary Fig. 12B). Therefore it seems that, instead of a single route that is coordinately regulated by Atg16L1 and A20, a complex mixture of signaling events involving NF- κ B activation, induction of unconventional autophagy and (probably) protection from cell death are activated in single-deficient cells to compensate for the absence of the relevant partner. In order to summarize all findings, we have now included a schematic graphical overview and a table in a new Supplementary Fig. 12 to summarize the most important conclusions which can be made in the different knockout conditions (cells and mice).

2b. Furthermore, the organoid result in Fig 4B is not convincing and both 4A and 4B results should be accompanied by densitometric analyses showing the reproducibility of this finding.

We now also included densitometric analysis for both Western blots. We have repeated the experiments shown in Figure 4 multiple times and consistently see subtle but enhanced expression of Atg16L1 and enhanced LC3 processing in A20 deficient cells. We moved the original Figure 4A to a new Supplementary Figure 4 (because of the air bubble as remarked by reviewer 2) and included a new representative figure in Figure 4 showing these differences in Atg16L1 expression and LC3 conversion. To convince the

reviewer of the reproducibility of this finding, a graph representing the densitometric analysis of 5 independent experiments on MEFs is shown in Figure 1 below (for referees only).

Figure 1. Graphs representing densitometric analysis of Atg16L1 expression and LC3-II expression (normalized to WT) measured on individual Western blots from 5 independent experiments on MEFs. Two-way ANOVA, multiple comparisons, with Bonferroni correction.

2c. *The data not shown on lack of ubiquitination of Atg16L1 and lack of stabilisation on proteasome inhibition should be included at least as Supplementary data because, although negative, these are important findings.*

We agree. As requested, we have now included a new Supplementary Fig. 6 showing that, while treatment with lactacystin induced substantial accumulation of ubiquitinated proteins in anti-UB (FK2) Western-blots performed in whole cell lysates, lactacystin did not stabilize transfected GST-ATG16L1 nor it favoured ubiquitination of immunoprecipitated GST-ATG16L1. Our interpretation of this result is that ATG16L1 is unlikely to be susceptible to physiological ubiquitination, suggesting that its stabilization in A20-deficient MEFs is probably caused by an alternative mechanism.

3a. *The phenotype in the IEC DKO mice is marked, but could be described in more detail. While the main emphasis is that there is enhanced apoptosis, in fact in vivo there is marked proliferation and even appears signs of dysplasia potentially in the SI. The histological phenotype requires more detailed description and addition of scoring of inflammation, crypt elongation etc across all regions of the small and large intestine. Clarity surrounding whether the primary defect is in secretory cells or affecting stem cells, enterocytes etc would aid interpretation.*

We show that spontaneous intestinal pathology in IEC-specific A20-Atg16L1dKO mice results from increased epithelial apoptosis and IEC hyperproliferation. In conditions of inflammatory pathology (as seen in IBD), epithelial apoptosis mostly occurs in proliferative crypt epithelial cells, hence epithelial apoptosis and hyperproliferation coincide and indeed often induce dysplasia.

We have now better characterized the histological phenotype by also including histology from the distal part (ileum) of the small intestine from the 4 genotypes (WT, A20 KO, Atg16L1 KO and A20-Atg16L1 dKO) (new Supplementary Figure 10 in the revised manuscript). Here we see similar pathology in the dKO mice as we describe in the Figure 6 for the jejunal part of the small intestine, *viz.* crypt elongation, villus blunting, presence of crypt abscesses, immune cell infiltration, reduced numbers of goblet and Paneth cells, epithelial hyperproliferation and apoptosis. We also quantified the pathology by doing histology on tissue sections (Supplementary Fig. 10B). The pathology in the small intestine is however milder in the distal ileum compared to the proximal jejunum, as shown in Figure 2 below (for referees only).

Figure 2. Histological scoring (left) and quantification of cleaved caspase 3-positive cells (right) comparing proximal and distal sections of small intestine in dKO mice. *, $p < 0,05$.

For the colon, based on colonoscopy, macroscopic and histological examinations, the biggest lesions are seen in the distal parts of the colon (for which histological pictures and quantification are shown in Figure 6 and Supplementary Figure 11 of the manuscript). Only minor pathology is observed in other parts of the colon (data not shown).

We now also analysed the expression of the stem cell marker *Lgr5* and the Paneth cell markers *Lysozyme P* and *cryptdin-1* in intestinal lysates from the 4 genotypes (WT, A20 KO, Atg16L1 KO and A20-Atg16L1 dKO) demonstrating significant reduction in expression of these markers in the dKO condition (new Figure panels Fig. 6I-J).

Finally, we now also included histology from small intestine and colon from 3-4 week old A20-Atg16L1 dKO mice (new Supplementary Figure 9). At this young age, these dKO mice already display severe intestinal defects, similar as we described for the older mice. However, based on these analyses it is hard to say what the primary defect is which induces the overall intestinal pathology. The villin-Cre transgenic line used to generate IEC-specific A20-Atg16L1 dKO mice targets all epithelial cell lineages of the intestinal tract, and expression mediates efficient Cre-mediated recombination in all IECs starting before birth (Madison et al., J. Biol. Chem., 2002). Nevertheless, secretory cells including Paneth and goblet cells are known to be particularly sensitive to cell death induced by inflammatory cytokines, ER-stress and autophagy (Vereecke et al., Trends Mol. Med., 2011), so barrier destabilization in dKO mice may be initiated by cell death of secretory cells. To specifically assess the effect of combined A20-Atg16L1 deletion in specific epithelial cell types on tissue homeostasis and the development of intestinal pathology, cell type-specific (stem cell, goblet cell, Paneth cell) approaches will be required, e.g. using cell-specific Cre lines, but given the long time required to generate such mice, we believe this is not feasible in the context of this manuscript.

3b. The lack of growth of the organoids from DKO mice is interesting but not fully explored – while anti-TNF did not work it would be good to explore chemokine release by the DKO epithelium and in fact how organoids would respond to the inflammatory environment established in the mice in vivo where hyperproliferation is the phenotype.

We tried many conditions (including incubation with apoptosis, necroptosis and ferroptosis inhibitors, see also below in reply to the comment of reviewer 2) in order to get viable dKO organoid cultures. Unfortunately, none of these conditions was able to protect A20-Atg16L1 dKO organoids from dying.

Besides the fact that expression of inflammatory cytokines (TNF, IL-1 β , CXCL2 and MCP1) can be detected in small intestinal lysates from dKO mice (as shown in Fig. 5J), we now also isolated and incubated small intestinal explants from A20-Atg16L1 dKO mice *ex vivo* and analyzed their supernatant for inflammatory cytokines. Stimulating WT and A20-/- organoids with explant supernatant resulted in swelling and death of only A20-/- organoids, indicating that the cytokines produced in the dKO intestinal tissue is sufficient to kill A20 deficient epithelium (Fig. 7C). We showed previously that A20 deficient organoids are hypersensitive to multiple cytotoxic cytokines,

including TNF and IFN- γ (Vereecke et al., Nat. Commun 2014). We therefore speculate that dKO organoids are hypersensitive to multiple toxic cytokines, either produced in an autocrine fashion or present during dKO organoid isolation. Since multiple cytokines (e.g. TNF, IFN- γ , IL-1 β) induce cytotoxicity through different cell death modalities, rescuing dKO organoids may not be possible by inhibiting individual cell death pathways.

Minor Points

1. *Figure legend 3 refers to IP methods in Fig 1 – this should be Fig 2.*

This mistake has been corrected in the revised version of the manuscript.

Reviewer #2:

Identification of interaction partners of ATG16L1 is of interest to the field. The authors demonstrate that A20 (besides a plethora of other proteins) and ATG16L1 physically interact in cells other than intestinal epithelium. The mechanism how A20 regulates ATG16L1 level is unclear but may be explained by work from Diamanti et al (JEM 2017). When the physical interaction between A20 and ATG16L1 fails due to deletion of both proteins in intestinal epithelial cells, spontaneous ileo-colitis develops which is associated with IEC death. These data suggest that A20 and ATG16L1 are critical to maintain IEC homeostasis.

We appreciate the reviewer's insightful comments and suggestions. In the following we provide a point-by-point reply.

1. *Whether cell death is a consequence of an interplay between A20 and ATG16L1 is unclear because the relevance of a direct physical interaction between A20 and ATG16L1 is unresolved. It appears that the presented protein studies are uncoupled from the notable phenotype they observe in DKO mice.*

We agree. The relevance of the direct physical interaction between A20 and ATG16L1 is important to interpret the mouse phenotype.

To try to address this point we reconstituted Atg16L1/A20-double deficient MEFs with retroviral constructs expressing HA-A20 and/or different versions of ATG16L1 (full-length or just the N-terminal domain (delta-WDD)). Initial experiments were directed to recapitulate the previously observed stabilization of endogenous Atg16L1 in the absence of A20 (Fig. 4A), in order to subsequently explore the requirement of the WDD in this process. However, such efforts were unsuccessful, perhaps because the subtle effect observed for endogenous Atg16L1 is lost when both A20 and ATG16L1 are expressed at higher levels by retroviral transduction.

However and unexpectedly, in these experiments we found that ATG16L1 causes destabilization of A20, since MEFs expressing transduced ATG16L1 showed reduced levels of A20 (particularly after treatment with TNF; see the new Figs. 4E and 4F). Such destabilization is apparently mediated by the WDD, because absence of this region recovered A20 expression, (Fig. 4E and 4F). Importantly, to generate this cellular system, cells were transduced first with HA-A20 and later with the different ATG16L1 constructs, so any change in A20 expression between the different strains cannot be caused by a dissimilar transduction efficiency of A20. Considering that A20 is expressed from a constitutive retroviral construct, the observed destabilization of A20 is likely a post-transcriptional event. In fact, treatment with lysosomal inhibitors (bafilomycin, E64d/pepstatin) tended to normalize A20 expression in cells harbouring different ATG16L1 status (no Atg16L1, full-length ATG16L1 or just the N-terminal domain; see the new Supplementary Fig. 7A), arguing that a lysosomal degradation pathway (likely autophagy) functions more intensely in the presence of full-length ATG16L1 to eliminate A20. In addition, we provide new data indicating that endogenous A20 is also regulated by ATG16L1 through the WDD. Thus, Atg16L1-

deficient cells treated with TNF showed increased levels of A20 associated with the appearance of higher molecular weight forms of this protein (new Supplementary Fig. 8A). Such increased expression of A20 is returned back to normal levels upon re-expression of full-length ATG16L1 but not in cells restored with just the N-terminal domain (Supplementary Fig. 8A), suggesting again that the WDD is critical for A20 destabilization by ATG16L1. Since the WDD mediates the direct interaction between A20 and ATG16L1, we believe that the A20 destabilizing activity of ATG16L1 likely requires direct binding between both molecules.

Given the important role of A20 in the control of NF- κ B activation induced by TNF, we measured the possible impact of the varying expression levels of A20 in the NF- κ B response using a suitable NF- κ B-luciferase reporter system. While we were unable to evaluate this issue in cells restored with HA-A20 (since overexpression of A20 dominantly inhibited NF- κ B activation in all strains; see the new Supplementary Fig. 7B), we found that, in the absence of A20, Atg16L1 reduces NF- κ B activation, because A20/Atg16L1-double deficient MEFs restored with ATG16L1 responded less efficiently to TNF treatment (Supplementary Fig. 7B). Again, the WDD seems to be important in this context, because cells expressing just the Nt domain of ATG16L1 behaved as Atg16L1-deficient cells (Supplementary Fig. 7B).

Similar studies in Atg16L1-deficient MEFs showed that absence of Atg16L1 increases NF- κ B responsiveness to TNF (new Supplementary Fig. 8B), suggesting again that Atg16L1 represses NF- κ B activation. Consistently, re-expression of ATG16L1 or just the N-terminal domain suppressed the signal, although the effect of the latter was less pronounced (Supplementary Fig. 8B). These data argue again that the WDD mediates at least in part the suppressive effects of Atg16L1 in this context. Interestingly, given that Atg16L1-deficient cells show increased A20 expression (Supplementary Fig. 8A) and considering that A20 is generally thought to repress NF- κ B activation in the TNF pathway, the NF- κ B inhibitory function of Atg16L1 is probably unrelated to its ability to control A20 expression levels. This notion is also sustained by the results in Supplementary Fig. 7B, where we show that transduced ATG16L1 inhibits NF- κ B activation by TNF in A20-deficient cells. Taken together, these results suggest that Atg16L1 regulates TNF-induced NF- κ B activation at least in part through its WDD and independently of A20.

We conducted a series of additional studies to explore the nature of the increased autophagic response observed in A20-deficient MEFs (as shown in Fig. 4A). In these studies we found that accumulation of LC3-II in these cells likely reflects a more active autophagic flux, since A20^{-/-} cells showed increased LC3-II levels even in the presence of the lysosomal inhibitor bafilomycin (see the new Supplementary Fig. 5A for basal autophagy, and Supplementary Fig. 5B for TNF-induced autophagy). However, the idea that A20-deficient cells have a hyperactive autophagic pathway is contradicted by data indicating a parallel accumulation of p62 (see Fig. 4A), which is normally interpreted a symptom of blunted autophagic flux. Interestingly, expression of the WDD in A20^{-/-} cells (via retroviral transduction) inhibited in a dominant-negative manner the accumulation of LC3-II induced by TNF (new Supplementary Fig. 5C), indicating that the autophagic response detected in these conditions has unconventional features (WDD-mediated) that could help explain the apparently contradicting behavior of LC3-II and p62 in this setting. Alternatively, p62 is an early NF- κ B response gene (Vadlamudi and Shin, *Febs Lett.*, 1998), and therefore likely to be overexpressed in A20 conditions.

Lastly, we provide new results showing induced interaction (co-precipitation) between endogenous A20 and transduced ATG16L1 in response to TNF both in MEFs and HCT116 cells (see the new Fig. 3C). In our view, these data help strengthen the notion that both molecules functionally interact through their physical association in response to physiological stimuli.

Therefore, taken together, all these results point to the idea that there is a tight interplay

between A20 and Atg16L1 where their direct interaction through the WDD controls each other's stability (see the new Supplementary Fig. 12A), thus providing a possible mechanistic basis for their genetic interaction in the control of intestinal homeostasis. Loss of one interaction partner results in a compensatory upregulation of the other, resulting in only sensitization but no spontaneous epithelial cell death (cfr. mild phenotype in single-deficient mice). In contrast, losing both interaction partners results in spontaneous enterocyte cell death and intestinal inflammation. Our results in MEFs show that induction of Atg16L1 in A20-deficient cells promotes unconventional (WDD-dependent) autophagy, enhanced p62 expression and likely reduction of the increased NF- κ B activation levels that occur in A20-deficient conditions (see summary in the new Supplementary Fig. 12B). Conversely, A20 upregulation in cells lacking Atg16L1 correlates with higher levels of p62, increased NF- κ B activation and, according to the abundant literature on the functional role of A20, probably increased protection against cell death (Supplementary Fig. 12B). Therefore, it seems that, instead of a single route that is coordinately regulated by Atg16L1 and A20, a complex mixture of signaling events involving NF- κ B activation, induction of unconventional autophagy and (probably) susceptibility to cell death are activated in single-deficient cells to compensate for the absence of the relevant partner.

In order to summarize all findings, we have now included a schematic graphical overview and a table in a new Supplementary Fig. 12 to summarize the most important conclusions which can be made in the different knockout conditions (cells and mice).

2. Relevant questions that would address a potential interplay between A20 and ATG16L1 have not been addressed in vitro or in vivo. For example, is A20 required for autophagy in IECs or do the authors observe induction of autophagy in IECs from A20-ko mice?

We agree with the reviewer that the interplay between A20 and Atg16L1 ideally needs to be addressed in IECs. Using organoid cultures from control and A20 deficient mice we confirmed the subtle but enhanced expression of Atg16L1 in A20 deficient cells (Fig. 4B and Supplementary Fig. 4B). We also tried many other approaches using mice (including GFP-LC3 transgenic mice which we crossed into the A20 deficient background) and organoid cultures, in order to provide mechanistic insights regarding this interaction. Unfortunately, due to technical issues, we were not able to convincingly confirm our findings in IECs using these approaches. First, we tried to image the LC3-GFP *in vivo* in mice either or not injected with TNF but encountered technical problems to identify specific signals due to background from dead cells, mucus, etc. and had resolution issues to resolve LC3-positive punctae. Next, we tried to image endogenous GFP in organoid cultures, but fixation procedures (which had been optimized) dimmed the fluorescence, and tried stainings using anti-GFP antibodies, as well as anti-LC3 antibodies, however without much improvement. Apart from dead cells, also here we had a lot of background from Paneth cell granules and from mucus. All this was done using a Zeiss LSM880 Laser Scanning Microscope with Airyscan technology, which allows to resolve structures of 120nm in xy-axis and 350nm in z-axis and which gives up to an 8-times improvement of signal-to-noise ratio, so using state-of-the-art microscopy. In conclusion, we tried various approaches without much success.

In figure 3 below (only meant for reviewers), we show one of the experiments we did trying to assess autophagy (by means of staining for LC3-positive punctae) in organoid cultures. However, no specific staining could be observed.

Figure 3. Organoid cultures from wild-type (WT) and A20^{IEC-KO} mice, either or not treated with TNF and/or bafilomycin and stained with anti-LC3 antibodies. Negative control refers to organoids stained with only secondary antibody, however also displays positive signal.

We hope we can convince the reviewer that we have done a lot of work to address his concern, but had no other choice but to use MEFs to do the relevant experiments.

Our new results in reconstituted MEFs provide evidence that is relevant to the general questions raised by the reviewer in this point. We found that A20 is important for the normal autophagic flux sustained by Atg16L1 or the N-terminal domain. Thus, p62 and LC3 Western-blot in the new Figs. 4E and 4F suggest that the basal autophagic flux is nicely restored in cells harboring HA-A20 and any of the two Atg16L1 constructs (increased LC3-II expression and decreased p62 levels), but not in their A20-deficient counterparts (which show restored LC3-II expression but increased p62 levels). Intriguingly, we also found an epistatic link between A20 and the WDD that seems to be important for the proper control of LC3-lipidation. Thus, as shown in Fig. 4E and 4F, LC3-II levels were substantially upregulated in A20-deficient cells expressing the N-terminal domain of Atg16L1, but this is not accompanied by enhanced degradation of p62. A less prominent but similar phenomenon is observed in HA-A20-expressing cells harbouring the N-terminal domain (increased LC3-II without parallel reduction in p62 expression; Fig. 4E). Therefore, LC3 lipidation occurring in the absence of A20 or the WDD in this cellular system does not seem to directly impact on the basal autophagic flux with a consequent increased clearance of p62. Exploring where and how this LC3 lipidation takes place requires further experimentation, but from these results it is clear that the WDD is important for a functional crosstalk between A20 and Atg16L1 in the proper control of autophagic pathways.

In addition, as mentioned above, we have studied in more depth the consequences of the increased levels of LC3-II and Atg16L1 observed in A20^{-/-} MEFs (see Fig. 4A). We found that accumulation of LC3-II in these cells (both basally and in response to TNF) likely reflects a more active autophagic flux, because the increased LC3-II levels are also observed in the presence of the lysosomal inhibitor bafilomycin (see the new Supplementary Figs. 5A and 5B). However, the idea that A20-deficient cells have an overactive autophagic pathway is apparently contradicted by results indicating a parallel accumulation of p62 (see Fig. 4A), a phenomenon that is normally interpreted a symptom of blunted autophagic flux. Interestingly, expression of the WDD in A20^{-/-} cells (via retroviral transduction) inhibited in a dominant-negative manner the accumulation of LC3-II induced by TNF (new Supplementary Fig. 5C), indicating that the autophagic response detected in these conditions has unconventional features (WDD-mediated) that could help explain the apparently contradicting behavior of LC3-II and p62 in this setting. Alternatively, p62 is an early NF- κ B response gene (Vadlamudi and Shin, *FEBS Lett.*, 1998), and therefore likely to be overexpressed in A20 conditions.

Taken together, all these results argue that A20 plays an unanticipated important role in

the regulation of autophagy.

3. Does ATG16L1-mediated autophagy degrade A20 to control NF- κ B signalling?

We agree that this is a very important point. As mentioned above, our new data in reconstituted MEFs show that ATG16L1 destabilizes A20 (new Figs. 4E and 4F and Supplementary Fig. 8A), suggesting a role of Atg16L1-mediated autophagy in A20 degradation. Consistently, treatment with lysosomal inhibitors (bafilomycin, E64d/pepstatin) tended to normalize A20 expression levels in all cellular strains irrespective of their ATG16L1 status (no expression, full-length ATG16L1 or just the N-terminal domain; see the new Supplementary Fig. 7A), raising the idea that a lysosomal degradation pathway (likely autophagy) is more intensely degrading A20 in the presence of ATG16L1. Unexpectedly, lysosomal inhibition stabilized A20 in all strains, and the reason for such effect is unknown (Supplementary Fig. 7A). Taken together, these results are consistent with a model where Atg16L1-mediated autophagy degrades A20, as suggested by the reviewer in this point. While the mechanistic bases of this phenomenon are unclear, it is worth mentioning that A20 destabilization requires the WDD, whereas maintenance of the basal autophagic flux (measured as p62 levels) does not (Fig. 4E and F). Therefore, A20 degradation is probably not mediated by a purely canonical autophagic route in this system. One possibility could be that ATG16L1 is somehow acting as a receptor for selective autophagy by recognizing A20 through the WDD and carrying it for autophagic degradation, but of course this is purely speculative and demonstration of such role would require substantial experimentation.

As suggested by the reviewer in this point, we examined the impact of ATG16L1 WDD-dependent degradation of A20 in NF- κ B activation by measuring NF- κ B-dependent luciferase activity induced by TNF in all restored cell lines. As mentioned above, in these assays we found that ATG16L1 regulates NF- κ B activation in cells lacking A20, because cells expressing FL-ATG16L1 were less responsive than those devoid of ATG16L1 expression or those harbouring just the N-terminal domain (see the new Supplementary Fig. 7B), suggesting that the WDD mediates (at least partially) NF- κ B inhibition in this context. However, all strains restored with HA-A20 showed poor NF- κ B activation irrespective of their ATG16L1 status, likely because A20 overexpression dominantly inhibits the activity (Supplementary Fig. 7B). Therefore, the influence of A20 degradation by ATG16L1 in NF- κ B signalling could not be measured in this system. We have also tried to assess susceptibility to TNF-induced cell death of the whole cell panel. However, none of the cell lines was susceptible to TNF toxicity, perhaps because they acquired resistance during the immortalization process. Incubation in the presence of cycloheximide allowed TNF to induce cell death, but no differences could be established between the cellular strains (data not shown).

We also measured NF- κ B activation by TNF in the Atg16L1-deficient MEFs shown in the new Supplementary Fig. 8A. These studies showed that absence of Atg16L1 increases NF- κ B responsiveness to TNF (new Supplementary Fig. 8B), suggesting again that Atg16L1 represses NF- κ B activation. Consistently, re-expression of ATG16L1 or just the N-terminal domain suppressed the signal, although the effect of the latter was less pronounced (Supplementary Fig. 8B). These data argue again that the WDD mediates at least in part the suppressive effects of Atg16L1 in NF- κ B activation. Interestingly, given that Atg16L1-deficient cells show increased A20 expression (Supplementary Fig. 8A) and considering that A20 is generally thought to repress NF- κ B activation, the inhibitory function of Atg16L1 in this context is unlikely to be related to its ability to control A20 expression levels. This notion is also sustained by the results in Supplementary Fig. 7B, where we show that transduced ATG16L1 inhibits NF- κ B activation by TNF in A20-deficient cells.

Taken together, these results suggest that Atg16L1 represses TNF-induced NF- κ B activation independently of A20, thus responding the specific question of the reviewer

in this point.

4. Which signalling event elicits intestinal inflammation? What is the mode of cell death? Is death cause or consequence of disease?

Recent evidence has clearly identified cell death to be a central feature in chronic inflammatory diseases, including inflammatory diseases of the gut. In the intestine, IBD can be triggered by epithelial cell death inducing barrier disruption and loss of Paneth and goblet cells producing protective anti-microbial peptides and mucus, respectively, allowing infiltration of luminal bacteria into the mucosa (Kondylis et al., Immunol. Rev., 2017). Both apoptosis and necroptosis have been associated with IBD (Pasparakis and Vandenabeele, Nature, 2015), and although apoptotic death is considered less inflammatory due to the containment of the cell content within apoptotic bodies, we believe that in our A20-Atg16L1 dKO model apoptosis is the driving cell death mode triggering chronic intestinal inflammation. We now included some discussion on this in the revised manuscript (page 13 of the revised manuscript). To further address the cell-death mode (apoptosis, necroptosis, ferroptosis,...) responsible for the phenotype of dKO mice, genetic crosses with knockouts of central cell death regulators could be useful (RIPK3, MLKL, RIPK1 kinase dead mutant,...) but given the long time required to generate these mice (requiring genetic deletion of 7 alleles), we believe this is not feasible in the context of this manuscript. Alternatively, we have attempted to answer this question using selective inhibitors on primary organoid cultures, but have unfortunately not been able to increase our insights in the cell death mode of dKO enterocytes. Possible shifts in cell death modes during selective inhibition of one mode (e.g. apoptosis to necroptosis shifts), may explain the difficulty of proving the responsible cell death modality *in vivo*. Given the clear observation of cleaved caspase-3 positive cells in dKO mice (Fig. 6K and Supplementary Figure 9, 10 and 11), we believe apoptosis is the main cell death modality induced by combined A20 and Atg16L1 deficiency.

Finally, in order to summarize all findings, we have now included a schematic graphical overview and a table in a new Supplementary Fig. 12 to summarize the most important conclusions which can be made in the different knockout conditions (cells and mice).

5. The authors utilise non-epithelial cell lines for their proteomics analysis, the interactome in IECs may be different.

We agree. In fact, our proteomics results indicate that the WDD interactome is quite different in different cell lines or activation status (Venn diagram; Fig. 1B). When setting up the procedure, we tested a number of cell lines for expression levels and stability of the GST-WDD protein, and HCT116 cells showed poorer results in both parameters, so we chose alternative cell lines to maximize the eventual productivity of the approach. While we agree with the reviewer that ATG16L1 and A20 could interact in other cell lines but not in IECs, in the revised manuscript we provide new results showing detectable co-immunoprecipitation between both molecules in the HCT116 colorectal epithelial cell line upon TNF treatment (as well as a similar result in MEFs), arguing that the ATG16L1 interactome in intestinal epithelial cells does include A20.

This new result (along with a positive co-precipitation result obtained in MEFs in similar conditions), are included as Fig. 3C in the revised manuscript.

6. Figure 4: LC3 conversion (indicative for autophagy induction) occurs at baseline but not after TNF stimulation (A). This finding may be important and requires proper immunoblots without blurred LC3 bands (bubbles?) and irregular B-Actin. In contrast A20-/- small intestinal epithelial organoids did not exhibit increased Atg16l1 expression or LC3 conversion at baseline. As such, the regulation in MEFs and IEC organoids appears different as it stands now. If the authors claim induction of autophagy in A20-/- cells, they must use more assays than an immunoblot of LC3 to proof this concept.

We agree with this reviewer that the difference in Atg16L1 expression and LC3 conversion in MEFs is already present at baseline. However, these differences get more pronounced after TNF stimulation. We have now repeated the experiments shown in Figure 4 multiple times and consistently see subtle but enhanced expression of Atg16L1 and enhanced LC3 processing in A20 deficient cells. We moved the original Fig. 4A to a new Supplementary Fig. 4 (because of the air bubble as remarked by the reviewer) and included a new representative figure in Figure 4 showing these differences in Atg16L1 expression and LC3 conversion.

To convince the reviewer of the reproducibility of these findings, a graph representing the densitometric analysis of 5 independent experiments on MEFs is shown in Figure 1 of this rebuttal (as shown above, for referees only).

As suggested by the reviewer (and also as mentioned above in the response to previous points), we have used additional assays to explore the nature of the increased autophagic response (enhanced levels of LC3-II) observed in A20-deficient MEFs. We now include new results showing that A20-deficient MEFs display increased autophagic flux, both basally (new Supplementary Fig. 5A) and upon TNF treatment (new Supplementary Fig. 5B), since in both cases the increased levels of LC3-II observed in A20-deficient compared to WT MEFs persist after lysosomal inhibition with bafilomycin. However, the mechanism involved in activating such flux may be unconventional, because the enhanced LC3-II signal is blunted in a dominant-negative manner by retroviral expression of the WDD (see the new Supplementary Fig. 5C). It is now clear from different publications that the WDD of ATG16L1 is unnecessary to sustain the basal autophagic flux regulated by the canonical route (see, for example: Boada-Romero et al. Nat. Commun. 2016; Fletcher et al. EMBOJ. 2018). Therefore, these results argue that A20-deficient cells react to the absence of A20 by enhancing their autophagic flux through unconventional mechanisms involving the WDD. The exact nature of this response is currently unclear and its detailed characterization is a very interesting issue that will need substantial experimentation. The involvement of unconventional mechanisms could explain why the behavior of LC3-II and p62 in A20-deficient MEFs (see Fig. 4A) seem to point to different conclusions (increased and blocked autophagic flux, respectively).

7. Figure 5A: Is Atg16l1 required for A20 expression or does it stabilise A20? Is its transcriptional Regulation of Atg16l1 by A20?

The anti-A20 Western-blot shown in Fig. 5A was meant to demonstrate absence of the relevant proteins in the respective KO organoids. Blots are accompanied by an irregular anti-Actin signal that complicates proper interpretation of the apparently decreasing A20 levels in Atg16L1-deficient cells.

However, the new data provided in the revised Fig. 4 (panels 4E and 4F) and the new Supplementary Fig. 8A indicate that ATG16L1 actually destabilizes A20, and it does so through its WDD in a post-transcriptional manner (see a more complete discussion around these issues in the points above).

We have now also measured mRNA levels of Atg16L1 in WT and A20 deficient MEFs and organoids. However, we do not detect a significant difference in Atg16L1 expression between both genotypes, concluding that the effect of A20 on Atg16L1 expression is regulated at the protein level. These data have now been included in the revised version of the manuscript (Figure 4C and D).

8. Does a-tocopherol or ferrostatin-1 protect against cell death of A20/Atg16l1-/- organoids?

We tried many conditions (including incubation with apoptosis and necroptosis inhibitors, as well as with the ferroptosis inhibitors a-tocopherol or ferrostatin-1, as suggested by the reviewer) in order to get viable dKO organoid cultures. Unfortunately, none of these was able to protect A20-Atg16L1 dKO organoids from dying. In addition to the data shown in Fig. 7 of the manuscript, a representative picture from these organoid cultures grown in presence of ferroptosis inhibitors is shown below

(Figure 3, for referees only).

Figure 3. Organoids from A20/Atg16L1 dKO animals fail to grow in culture. Pictures of wild-type (WT) and dKO intestinal organoids that were left untreated or were cultured in the presence of ferrostatin (500 nM) or a-tocopherol (100 μ M) for 24 h or 6 days. Pictures representative of 3 independent experiments.

Besides the fact that expression of inflammatory cytokines (TNF, IL-1 β , CXCL2 and MCP1) can be detected in small intestinal lysates from dKO mice (as shown in Fig. 5J), we now also isolated and incubated small intestinal explants from A20-Atg16L1 dKO mice *ex vivo* and analyzed their supernatant for inflammatory cytokines. Stimulating WT and A20-/- organoids with explant supernatant resulted in swelling and death of only A20-/- organoids, indicating that the cytokines produced in the dKO intestinal tissue is sufficient to kill A20 deficient epithelium (Fig. 7C). We showed previously that A20 deficient organoids are hypersensitive to multiple cytotoxic cytokines, including TNF and IFN- γ (Verecke et al., Nat. Commun 2014). We therefore speculate that dKO organoids are hypersensitive to multiple toxic cytokines, either produced in an autocrine fashion or present during dKO organoid isolation. Since multiple cytokines (e.g. TNF, IFN- γ , IL-1 β) induce cytotoxicity through different cell death modalities, rescuing dKO organoids may not be possible by inhibiting individual cell death pathways.

9. *As it stands now, the title is overstated as it implies a direct link between a common pathway controlled by A20 and ATG16L1 that instigates intestinal inflammation. It may be that the mere lack of both proteins induces death of IECs due to important homeostatic (independent) functions of both proteins.*

We agree. We have now changed the title of the manuscript to a less direct version: “Physical and functional interaction between A20 and ATG16L1-WD40 domain in the control of intestinal homeostasis”

Discussion:

10. *The notion that “Atg16L1 deficiency in IECs does not cause spontaneous Paneth cell defects” is wrong. (see work from Cadwell, Adolph and Conwell et al.)*

We changed the text ‘... Paneth or goblet cell defects’ to ‘... Paneth or goblet cell death’, since in our studies no spontaneous Paneth or goblet cell death is observed in single IEC-specific Atg16L1 knockout mice.

11. *“Our data showing that the combined deletion of A20 and Atg16L1 provokes loss of Paneth and goblet cells sustains the idea that defects in the innate defense mechanisms linked to these cells contribute to the spontaneous development of intestinal pathology. These data are in line with previous work, showing that loss of either regulators of ER stress or modulators of autophagy result in each other’s compensatory engagement and that spontaneous pathology only develops in case both mechanisms are defective (Adolph et al*

2013).” *The referenced study in this context does not support the notion that dying IECs such as Paneth cells trigger intestinal inflammation. This study demonstrated that stressed Paneth cells (that do not die) elicit intestinal inflammation. Other studies indeed demonstrated that dying IECs (and the absence of Paneth or goblet cells) elicit a barrier defect and consequently intestinal inflammation (Gunter C Nature 2008).*

We made some changes to the text, based on the reviewer’s suggestion.

REVIEWERS' COMMENTS:

Reviewer #1 (Remarks to the Author):

The authors have responded to the critiques provided by myself and the other reviewer with new experiments and a very detailed response. These new experiments have provided more clarity and some unexpected findings which considerably strengthen the manuscript. These data are included in several of the main figures and in Supplementary figures. The manuscript has been revised accordingly and I am happy to accept the interpretations made concerning the new data. Some other experiments were attempted that proved technically challenging, and this has been explained appropriately in the response.

I have no further suggestions to make.

Michael McGuckin

Reviewer #2 (Remarks to the Author):

The authors have made substantial efforts to address the reviewers concerns and improved the study by providing more experimental information and revising the manuscript. The findings are very interesting and this reviewer hopes to see more mechanistic insights that may explain the phenotype in A20/Atg16l1 DKO mice in the future.